



# Optimized shaped pulses for 2D-SIFTER

Paul A. S. Trenkler[1], Burkhard Endeward[1], Snorri Th. Sigurdsson[2], Thomas F. Prisner[1]

[1]Institute of Physical and Theoretical Chemistry and Center of Biomolecular Magnetic Resonance, Goethe University Frankfurt, Frankfurt am Main, 60438, Germany

[2]Science Institute, University of Iceland, Reykjavik, 107, Iceland

*Correspondence to*: Thomas F. Prisner (prisner@chemie.uni-frankfurt.de)

**Abstract.** Fast and accurate arbitrary waveform generators (AWG) for generating shaped pulses in EPR have been commercially available for over a decade now. However, while the use of chirp pulses as inversion pulses in pulsed electron double resonance (PELDOR) experiments has become common, their application for generating broadband phase-sensitive

transverse magnetization is not widely adopted within the community. Here we give a detailed insight into optimization procedures and instrumental challenges when using chirped pulses for broadband Fourier transform (FT) detection of electron spin echo signals, particularly the two-dimensional frequency-correlated single frequency technique for refocusing (SIFTER) experiment. To better understand the influence of chirped pulses on the generation of broadband transverse magnetization, we investigated the phase and amplitude of chirped echoes for different time-bandwidth products while varying the number of

refocusing pulses, particularly under the influence of $B_1$-inhomogeneity. Following our optimization procedures, we were able to perform orientation selective SIFTER measurements using rigid nitroxide spin labels on an RNA duplex. Finally, we also demonstrate the first experiments with two novel SIFTER pulse sequences, which could be of interest for the detection of either shorter or longer distances.

## 1  Introduction

Electron paramagnetic resonance (EPR) spectroscopy and nuclear magnetic resonance (NMR) spectroscopy are fundamentally rooted in the same spin physics and advancements in one field have historically inspired and benefited the other. Since the electron spin's magnetic moment is almost three orders of magnitude larger than that of the nuclear spin with the largest magnetic moment ([1]H), EPR offers intrinsically much higher sensitivity compared to NMR. However, the large difference in magnetic moment also results in significantly faster spin relaxation and much larger spectral widths, which have posed

longstanding technical challenges in pulsed EPR spectroscopy. The rapid relaxation of electron spins in paramagnetic molecules necessitates the use of cryogenic temperatures for most pulsed EPR experiments, where relaxation times are sufficiently prolonged to allow coherent spin manipulation and detection. Yet, at low temperatures, inhomogeneous line broadening by anisotropic hyperfine interactions and g-tensors becomes more pronounced, making uniform excitation over the entire spectral range impossible with rectangular microwave pulses. Thus, spectrally resolved hyperfine or dipolar



couplings have mostly been collected in a serial manner by varying the microwave (MW) detection pulse frequency or the external magnetic field to excite different parts of the EPR spectra.

One of the most popular pulsed EPR techniques is the pulsed electron-electron double resonance (PELDOR) experiment (Milov et al., 1981) also referred to as double electron-electron resonance (DEER) (Larsen and Singel, 1993). PELDOR and other pulsed dipolar spectroscopy (PDS) methods allow the determination of distances between two spin labels up to 16 nm

(Endeward et al., 2022; Kunjir et al., 2013; Schmidt et al., 2016). This distance regime is of interest for determining/investigating tertiary structure and conformational dynamics of biological macromolecules and complexes. Although other PDS techniques such as double quantum coherence (DQC) (Borbat and Freed, 1999) and single frequency technique for refocusing (SIFTER) (Jeschke et al., 2000) have been introduced, they have found less applications as of yet. The main reason for the more frequent use of PELDOR is that it relies on selective pulses and does not require excitation over

large bandwidths, whereas SIFTER and DQC rely on the excitation of the full spectral width. The most common spin labels in EPR are nitroxides and rectangular pulses are not capable of exciting the entire nitroxide spectrum, even at X-band frequencies. Therefore, broadband shaped-pulses are required to achieve full excitation of nitroxide radicals (Schöps et al., 2015). Although fast arbitrary waveform generators (AWGs) have been commercially available for over a decade, their integration into existing spectrometers, originally designed for rectangular pulses, is not straightforward. Amplifiers are

typically driven close to saturation, to increase their amplitude stability for rectangular pulses. However, for shaped pulses it is essential to keep all amplifiers in their linear regime, to preserve the designed pulse shape at all power levels. Numerous other spectrometer components, such as the resonator, video amplifier, transient recorder and the AWG itself have to be calibrated and optimized for the target bandwidth of the shaped pulse applications. Incomplete excitation by the microwave pulses creates additional unwanted signals in the time traces of SIFTER and DQC experiments, resulting from different

coherence pathways. These unwanted signals can impede quantitative data analysis. Until recently (Vanas et al., 2023) the background contribution to SIFTER had been largely unexplored and easy-to-use tools to perform this background correction have not been introduced while a number of tools for the background correction and analysis of PELDOR time traces exist (Abdullin et al., 2024; Fábregas Ibáñez et al., 2020; Jeschke et al., 2006; Worswick et al., 2018).

The majority of shaped pulses for broadband excitation used in EPR so far are chirped fast-passage pulses (Doll et al., 2013;

Spindler et al., 2013). The spin dynamics occurring during such pulses is different from rectangular pulses. With chirped excitation pulses, different offset frequencies are excited at different points in time, resulting in an offset-depended phase roll that has to be compensated by other pulses in the sequence. In NMR, this phase roll is sometimes compensated in post processing of the data (O'Dell and Schurko, 2008). This is not possible in EPR since the spectra are much broader, which results in destructive interference between different spin packets and loss of signal immediately due to this phase roll. Although

the spin response to shaped pulses for broad spectra and with inhomogeneous $B_1$ fields has been extensively studied in the field of NMR, similar studies are sparse for EPR applications (Garwood and DelaBarre, 2001).

In this work, we give new insights into the use of shaped pulses for broadband excitation and showcase how they can be used to get EPR-correlated dipolar time traces by 2D-SIFTER when applied to rigid spin labels. First, we provide a brief theoretical





background that is required for discussing the effects of chirped pulses on the spin system. Then we explore the concepts of chirped pulses for broadband excitation and, in this context, present established and novel simulation-driven insights into the effects of different pulse parameters on the echo phase and amplitude, particularly under $B_1$-inhomogeneity. We provide a detailed explanation on how to set up 2D-SIFTER experiments at X-band frequencies and discuss a two-step correction method for obtaining a spectrum by Fourier transformation (FT) of the SIFTER echo that matches very well with the echo detected field sweep (EDFS) spectrum. We then compare our 2D-SIFTER results of a doubly labelled RNA duplex with previously published orientation-selective PELDOR data on the same sample that was acquired using Gaussian-shaped pump and probe pulses (Gauger et al., 2024). Lastly, in an outlook section, we show initial experiments with two novel SIFTER sequences, one of which could be used for shorter distances and the other for longer distances. The first pulse sequence employs different pulse length ratios to minimize the frequency dispersion of the dipolar traces. The second sequence makes use of dynamic decoupling to minimize the transverse relaxation effects during the experiment and outperforms the classical 4 pulse-SIFTER in our first test measurements.

## 2 Chirped pulse echoes

In this section we first give a brief overview (Sect. 2.1) of the basic theoretical aspects of shaped pulses that are relevant to understand our discussions in Sect. 2.2. For a more detailed discussion on these aspects we refer to the literature (Endeward et al., 2023; Jeschke et al., 2015; Spindler et al., 2017; Verstraete et al., 2021). In Sect. 2.2 we explore, assisted by Bloch vector simulations, the different phase effects of chirped pulses and how they affect the chirp echo amplitude under $B_1$-inhomogeneity.

### 2.1 Important chirp pulse parameters

Chirped or fast-passage pulses can be fully described by an amplitude function $\omega_1(t)$ and its phase function $\phi(t)$ in the rotating frame in the following manner:

$$\boldsymbol{S(t) = \omega_1(t) \cdot \exp(i\phi(t))} \tag{1}$$

In this work, we only show data from shaped pulses with a linear frequency chirp and a WURST (wide band uniform rate smooth truncation) amplitude function (Kupce and Freeman, 1995). The instantaneous frequency function $\omega(t)$ and amplitude function $\omega_1(t)$ of a WURST pulse are given by the following equations:

$$\boldsymbol{\omega_1(t) = 1 - \left| \sin\left(\frac{\pi t}{t_p}\right) \right|^n}, \tag{2}$$

$$\boldsymbol{\omega(t) = \frac{2\pi \cdot SW}{t_p} \, t + \omega_c} \tag{3}$$

Here, $n$ is a dimensionless tuning parameter, which determines the steepness of the WURST truncation function, $SW$ is the sweep width of the linear frequency chirp, which is generally larger than the real excitation bandwidth $BW$, $\omega_c$ is the



microwave carrier frequency and $t$ is defined from $-t_p/2$ to $t_p/2$. We further define the time-dependent frequency offset $\Delta\omega(t)$ as the difference between the instantaneous frequency function and the center frequency $\Delta\omega(t) = \omega(t) - \omega_c$. For linear

frequency chirps in particular, this frequency offset is often more intuitive to interpret than the phase and the two are directly related by integration over the full pulse duration:

$$\phi(t) = \int_{-t_p/2}^{t} \Delta\omega(t')dt' \tag{4}$$

There are also two additional parameters that are of importance for WURST pulses. Those are the so-called critical adiabaticity $Q_{crit}$ and the time-bandwidth product ($TBP$).

$$Q_{crit} = \frac{\omega_1^2 \cdot t_p}{SW \cdot 2\pi} \tag{5}$$

$$TBP = t_p \cdot SW \tag{6}$$

$Q_{crit}$ is of particular importance, because it can be used to calculate the effective flip angle $\beta$ of the pulse (Jeschke et al., 2015).

$$\beta = \cos^{-1}\left(2 \cdot \exp\left(-\frac{\pi Q_{crit}}{2}\right) - 1\right) \tag{7}$$

With Eq. (7) one can calculate that a chirped $\pi/2$-pulse should have a value of $Q_{crit} \approx 0.44$. For a given pulse length $t_p$ and

sweep width $SW$ one can directly calculate the necessary amplitude $\omega_1$ and vice versa. The amplitude of a $\pi$-pulse is not so well-defined. The flip angle of chirp pulses, asymptotically approaches $\pi$ for higher values of $Q_{crit}$ and, therefore, a chirp pulse can flip neither polarization nor coherence by an angle larger than $\pi$ (Endeward et al., 2023). It has been mentioned that a value of $Q_{crit} = 5$ is generally sufficient (Baum et al., 1985). We show in Sect. 2.2.4 that in some cases, even smaller values can lead to larger echo signals.

The $TBP$ needs to be high enough to achieve a broad and smooth excitation without distortions. The two parameters $Q_{crit}$ and $TBP$ can counteract each other; it can be necessary to decrease the critical adiabaticity $Q_{crit}$ by sweeping over a larger $SW$ to achieve a sufficiently high $TBP$ (Verstraete et al., 2021). As a rough guideline one should try to avoid $TBPs$ with values lower than 30, but it is best to verify this by simulating the excitation profiles of all used pulses.

## 2.2 Using chirped pulses for the generation of broadband transverse magnetization

When using chirped pulses to generate and refocus transverse magnetization a few different phase effects need to be considered; we explore these by Bloch vector simulations and discuss them in this section. We give a brief description of our simulation procedure in Sect. 2.2.1. Then we discuss pulse-length ratios for refocusing spins that are brought into the transverse plane at different times and introduce a MATLAB script for finding ratios with minimal overall pulse length for a given coherence pathway (Sect. 2.2.2). Finally, we discuss phase effects that depend on $Q_{crit}$ in Sect. 2.2.3 and how they influence

the echo amplitude under $B_1$-inhomogeneity in Sect. 2.2.4.





### 2.2.1     Bloch vector simulations

All Bloch vector simulations were performed using a home-written MATLAB routine by numerically solving the Bloch equations by rotating spins with different offsets around the effective magnetic field, as described by Spindler et al. (2016). All simulations were performed with a set of 1000 independent spins $S = 1/2$ with an offset range of -100 to +100 MHz and without considering relaxation. In the simulations where $B_1$-inhomogeneity was considered (Fig. 3 and 4), we did so by performing multiple simulations. For each simulation the pulse amplitudes were scaled by the different $B_1$-strengths and the magnetization of all simulations was added together and weighted by the $B_1$-weights of the assumed distribution (shown in Fig. 3). The amplitude of the $\pi/2$-pulse was always set such that Eq. (7) gives a flip angle of $\beta = \pi/2$ and the same values of $Q_{crit}$ were applied to all $\pi$-pulses in multi-pulse experiments. All simulations were normalized to the number of spins and to the sum of the $B_1$-weights. This means that a fidelity value of 1 corresponds to all spins being aligned in phase at the echo time.

### 2.2.2     Pulse length ratios

It is important to realize that while a single chirped $\pi/2$ pulse can bring magnetization over a broad frequency range into the transverse plane, it does so by introducing a parabolic phase roll over its excitation bandwidth. This originates from the fact that spins at different offsets are brought into the transverse plane at different points in time. To distinguish this phase roll from other phase effects that we will discuss later, we gave it the symbol $\phi_p$. The parabolic phase roll $\phi_p$ can be refocused by additional pulses of different length and/or sweep direction. Jeschke et al. (2015) showed that for a pulse sequence of chirp pulses with the same sweep width $SW$, one can find a combination of pulse length ratio and sweep directions for a given coherence order pathway by satisfying the following condition:

$$\sum_i s_i t_{p,i}(o_i^- - o_i^+) = 0 \tag{8}$$

Here $s$ is a sign factor that is $+1$ for an upward sweep and $-1$ for a downward sweep, $t_p$ is the pulse length, $o^-$ is the coherence order before and $o^+$ the coherence order after the pulse. The pulse number is given by the index $i$. With this condition, one can find multiple solutions, depending on the number of coherence pathways and number of pulses. For a specific set of sweep directions, Eq. (8) will be a linear equation or a coupled linear equation system if multiple coherence pathways need to be considered. In principle, as long as the sequence has more pulses than coherence pathways, the linear equation system is underdetermined and has infinite solutions. Since in a real experiment only the relative pulse lengths are relevant, one further degree of freedom is removed and a sequence with two pulses and one coherence pathway will have one unique solution. This can be seen for the Hahn echo, where the unique solution for a sequence of pulses with only upward sweeps, is that the first pulse must be twice as long as the second pulse. This is known in NMR spectroscopy as the Böhlen-Bodenhausen or Kunz-Bodenhausen sequence (Bohlen et al., 1989; Kunz, 1986). Some sequences will not be underdetermined but have no meaningful solutions. For example, if one chooses to have an upward and a downward sweep in



the Hahn echo sequence for the first and second pulse respectively, the linear equation system will contain only negative numbers, so that negative pulse lengths would be necessary to obtain a solution. It should also be noted that if multiple coherence pathways are considered, a solution of Eq. (8) will refocus the magnetization of different offsets for all pathways but not necessarily at the same point in time (we discuss such a case in Sect. 4.2.1).

If a sequence has multiple solutions (which will be the case for most sequences) one can then choose a solution where the distribution of pulse lengths is advantageous for the specific experiment. Typically, one does not want to have a sequence where the relative length of a $\pi$-pulse is shorter than the length of a $\pi/2$-pulse, since the $\pi$-pulses generally require a much higher amplitude $\omega_1$. This means the shortest $\pi$-pulses in a sequence determines the minimal length one can have for the pulses of the sequence. The highest achievable amplitude $\omega_1$ (determined by the linear regime of the MW amplifier) determines the length of this pulse to achieve a sufficient high value of $Q_{crit}$. In cases where the relative length of a $\pi/2$-pulse is much shorter than the $\pi$-pulses, it might be the $TBP$ of this $\pi/2$-pulse that limits how short the pulses can be made.

We have written a MATLAB script which generalizes Eq. (8) to give a selection of solutions that are sorted by how short the sequence can be made for any desired pulse sequence. This approach automatically finds already known solutions such as the Böhlen-Bodenhausen sequence for the Hahn echo (Bohlen et al., 1989), the compressed CHORUS sequence for the refocused echo (Foroozandeh et al., 2019) and the ABSTRUSE equivalent of the stimulated echo (Jeschke et al., 2015) and can be used for any other pulse sequence. More details are given in the Appendix in Sect. B1 and the code is made publicly available.

### 2.2.3 The chirp echo phase

Another aspect that is very different for chirped pulses, compared to rectangular pulses, is the phase of the echo. In the case of rectangular pulses, the phase of the echo compared to the phase of the pulses is fixed and independent of the pulse lengths and amplitudes. For a Hahn echo with rectangular pulses along the x-direction, the echo will form along the y-direction. In the case of chirped pulses, this is different even when the pulse length ratios are set up according to Eq. (8). The echo phase is highly dependent on $Q_{crit}$. This effect has been explained by the Bloch-Siegert phase (Emsley and Bodenhausen, 1990) and has been discussed both in the context of NMR and EPR with chirped pulses (Cano et al., 2002; Jeschke et al., 2015). Here we want to distinguish between two phase effects. The change of the echo phase which is independent of the frequency offset $\Omega$ (and which is most conveniently studied at $\Omega = 0$) we term $\phi_0$ and an additional phase shift, which is dependent on $\Omega$, and termed $\Delta\phi$. To refer to both phase shifts, we use the term dynamic phase shifts; the symbols and the nomenclature are consistent with those used by Jeschke et al (2015). It is important to realize, that while the focus of the study by Jeschke et al. was on the offset dependent phase change $\Delta\phi$, the phase at the center of the frequency chirp $\phi_0$ is also highly dependent on $Q_{crit}$. For pulse sequences where the only goal is the generation of a detectable echo, this is not an issue since the constant phase offset can be corrected after detection by zero-order phase correction. For pulse sequences where the phase of the magnetization within the pulse sequence is relevant, for example during the application of a further pulse, this additional phase shift becomes very important. The SIFTER experiment and the DQC experiment are examples of such pulse sequences. For the SIFTER sequence





specifically, the echo that is formed by the first and second pulse should refocus along the y-axis (assuming the pulses are applied along the x-axis). In that case the $(\pi/2)_y$-pulse can invert the sign of the anti-phase term, which will lead to the refocusing of the dipolar coupling contribution. Thus, for SIFTER the value of $\phi_0$ at the time of the third pulse is important (we discuss the influence of this phase on the SIFTER sequence in more detail in Sect. 3.1). Also, in the presence of $B_1$-inhomogeneity, different values of $\phi_0$ over the sample can lead to destructive interference. This is discussed in more detail in Sect. 2.2.4.

Jeschke et al. (2015) realized by numerical simulation that there is a near linear dependence of both $\Delta\phi$ and $\phi_0$ on $Q_{crit}$ and both change sign if the sweep direction of a pulse is inverted. They mostly studied the offset dependent phase $\Delta\phi$ by applying zero-order phase correction to the echo of two chirped pulses with relative lengths of 2 to 1.

We performed numerical Bloch vector simulations to investigate the dependence of $\phi_0$ and $\Delta\phi$ on the pulse parameters sweep width $SW$, pulse length $t_p$ and time-bandwidth product $TBP$. This is illustrated in Fig. 1 (a) to (c) where, respectively, one of these three parameters was kept constant while the other two were changed. At sufficiently high $Q_{crit}$ we confirmed the results by Jeschke et al. and also found a near-linear dependence of $\phi_0$ on $Q_{crit}$. Additionally, we found that the slope of this linear dependence does not only depend on the $SW$ but also on the pulse length $t_p$. More accurately, it depends on the time-bandwidth product $TBP$. One can see that both pulse length and sweep width change the slope of $\phi_0(Q_{crit})$ (Fig. 1 (a-b)) but if both parameters are changed simultaneously, keeping the $TBP$ constant (Fig. 1 (c)), the slope of $\phi_0(Q_{crit})$ is constant.



In Fig. 1 (d) to (f) one can see that generally the phase shift of $\phi_0$ is very large compared to the effect of $\Delta\phi$ (note that contributions from $\Delta\phi$ become more dominant at the edges of the $SW$, as was shown by Jeschke et al. (2015)). In contrast to $\phi_0$, $\Delta\phi$ seems to only depend on the sweep width and not on the pulse length or the time-bandwidth product.

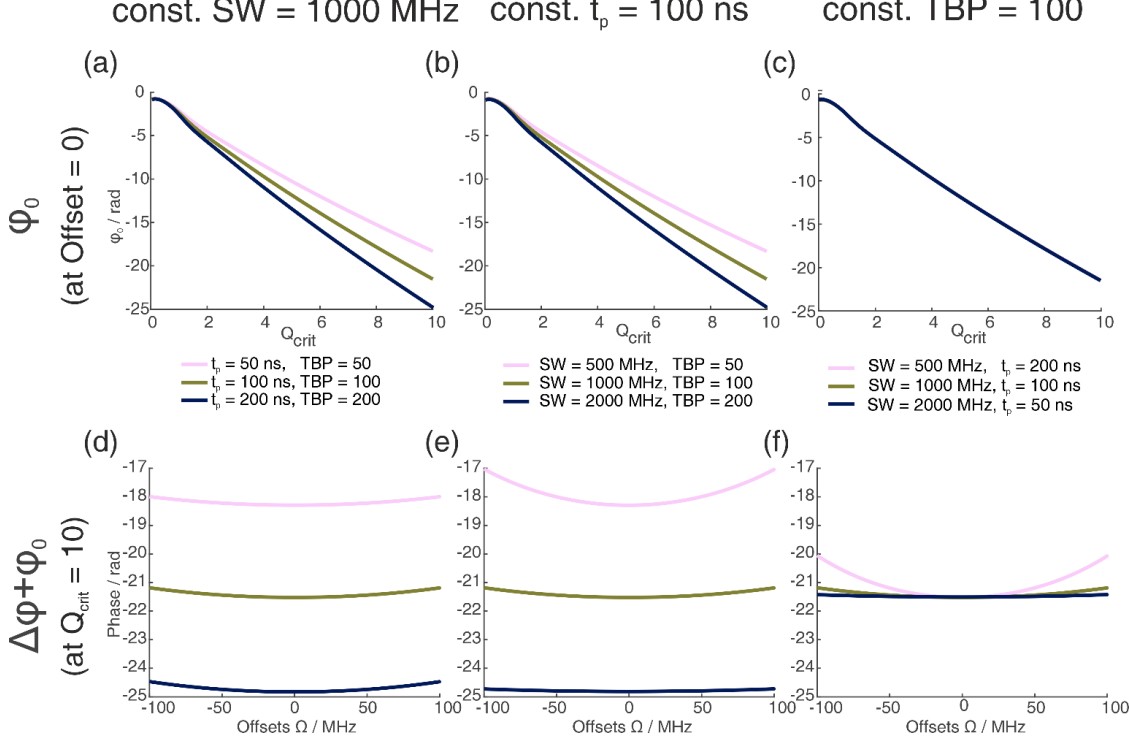

**Figure 1: Numerical Bloch simulations of two pulse chirp Hahn echoes with pulse length ratio 2-1. The top row ((a) to (c)) shows the phase $\phi_0$ at offset $\Omega = 0$ MHz as a function of $Q_{crit}$ of the second pulse and the bottom row ((d) to (f)) shows the combination of $\phi_0$ and $\Delta\phi$ at $Q_{crit} = 10$ as a function of $\Omega$. For each column ((a) and (d), (b) and (e), (c) and (f)) either the sweep width $SW$, the pulse length of the last pulse $t_p$ or the time bandwidth product $TBP$ are held constant while the other two parameters are varied. The nonlinear behaviour at low values of $Q_{crit}$ stems from incomplete coherence inversion and will not be further considered here.**

When observing $\phi_0$ and $\Delta\phi$ with a refocused echo sequence, or any other sequence with an even number of refocusing pulses, $\Delta\phi$ is refocused and $\phi_0$ is mostly refocused. This is because two $\pi$-pulses with the same value of $Q_{crit}$ can compensate their

respective dynamic phase shifts, if they both apply the same phase shift. In fact, for the refocused echo sequence with a pulse length ratio of 2-2-1, which has been given the acronym ABSTRUSE by Cano et al. (2002), the authors have shown that this sequence largely refocuses all dynamic phase shifts. In Fig. B1 of the appendix one can see that for the ABSTRUSE echo there is no remaining dependence of $\Delta\phi$ on the sweep width, pulse length or time-bandwidth product. However, there still is a small but significant dependence of $\phi_0$ on $Q_{crit}$ (compared to the Böhlen-Bodenhausen Hahn echo sequence). The remaining

dependence of $\phi_0$ on $Q_{crit}$ can be explained by the different relative pulse lengths of the two $\pi$-pulses used in this sequence. To refocus the parabolic phase roll $\phi_p$ of the first $\pi/2$-pulse, the ABSTRUSE echo applies a Böhlen-Bodenhausen-like sequence of with a pulse length ratio of 2-2-1 with equal sweep widths. It should be noted that in this sequence, the time-





bandwidth products of the two $\pi$-pulses are not equal. Therefore, they will each introduce a different phase shift $\phi_0$ and will not compensate their respective phase shifts. This is illustrated in Fig. 2, where we simulated two refocused echoes. In one

case the parabolic phase roll $\phi_p$ was refocused but the last two pulses had a different time-bandwidth product (2-2-1 sequence, black) and in the second case $\phi_p$ is not refocused but the two $\pi$-pulses have the same $TBP$ (2-2-2 sequence, red). For the latter, we observe no dependence of $\phi_0$ on $Q_{crit}$ at sufficiently high $Q_{crit}$ (Fig. 2 (a)), where effects of incomplete coherence inversion are no longer relevant. Since the parabolic phase shift is not refocused in this case, one can observe an offset-dependent phase shift in Fig. 2 (b) (red). While in case of the 2-2-1 sequence (black) there is a dependence of $\phi_0$ on

$Q_{crit}$, even at high values of $Q_{crit}$, but there is no offset dependent phase shift.

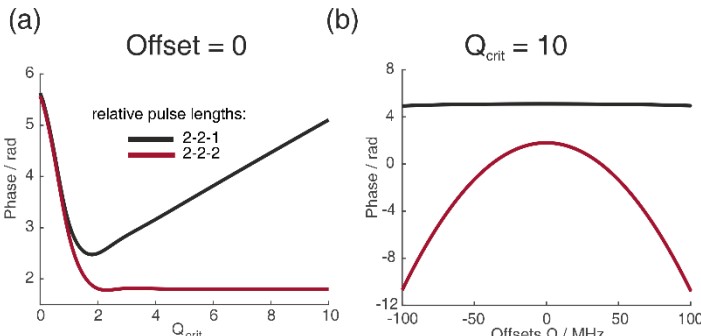

**Figure 2: Numerical Bloch simulations of chirped refocused echoes with pulse length ratios 2-2-1 and 2-2-2 with $t_p = 100$ ns. The sweep width was set to $SW = 500$ MHz. (a) shows the phase of the center of the echo at offset $\Omega = 0$ MHz as a function of $Q_{crit}$ of the $\pi$-pulses.**

### 2.2.4 The chirp echo amplitude

The flip angle $\beta$ of fast-passage pulses asymptotically approaches $\beta = \pi$ when increasing the amplitude of the pulse (see Sect. 2.1). Therefore, one could assume that in a 2-pulse Hahn echo experiment it is best to set up the $\pi$-pulse with the highest

achievable amplitude $\omega_1$. In reality, the echo intensity has a maximum at a specific value of $\omega_1$. This effect has been explained in the literature by the variance of $\Delta\phi$ at high $Q_{crit}$ (Cano et al., 2002; Jeschke et al., 2015). Both authors also mentioned that $B_1$-inhomogeneity might lead to a steeper decay of the echo amplitude. In most cases, considering only the dynamic phase shift $\Delta\phi$, it is not enough to explain the steepness with which the experimental chirped echo intensity decreases when the amplitude of the $\pi$-pulse is increased past its optimal value, as shown in Fig. 3. In Fig. 3 we have simulated the echo intensity

of a 2-pulse chirp Hahn echo experiment as a function of the amplitude of the $\pi$-pulse for different $B_1$-distributions. We assumed a half Gaussian distribution with different widths, since in a typical MW resonator, most of the sample will be exposed to the maximum $B_1$ field strength and only the parts of the sample at both outer edges of the MW resonator will experience a lower $B_1$ field strength.





We compared our simulations with experiments on two different samples. Both samples consist of TEIO-N nitroxide radicals
immobilized in trehalose (sample details in appendix A2). One sample was 4 mm long and squished between two paper towels
to keep the sample from the lower edge of the sample tube, ensuring a homogenous dielectric constant over the sample (black
in Fig. 3 (b)) and a second sample which extends beyond the resonator length (red). From Fig. 3 it is apparent that for broad
$B_1$ distributions one will have to accept a lower value of $Q_{crit}$ to achieve the maximum echo intensity. The half Gaussian

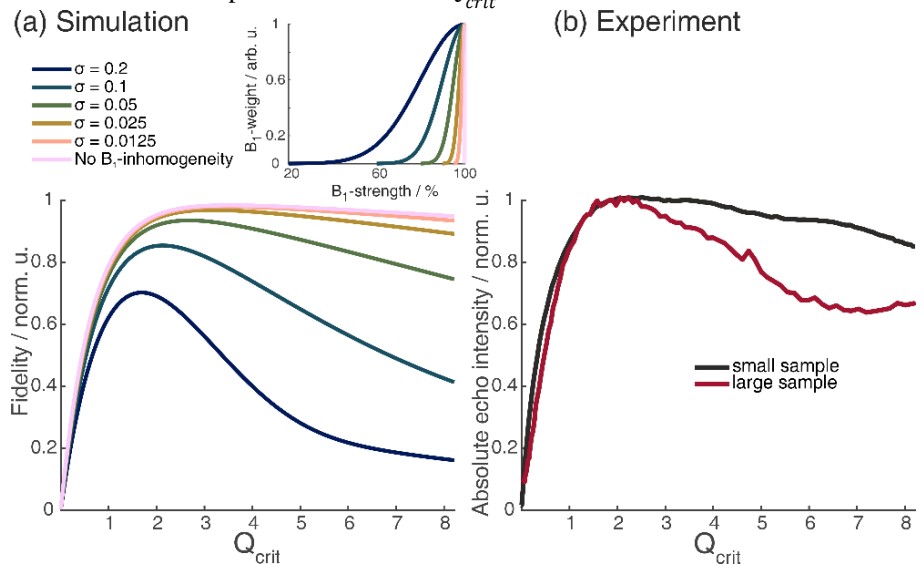

**Figure 3: Absolute echo signal intensity as a function of $Q_{crit}$ of the $\pi$-pulse in a chirped Hahn echo experiment. (a) Incremental Bloch vector simulation for a rectangular box of independent spins with an offset range of 200 MHz with two shaped pulses of lengths 200 ns and 100 ns, a linear frequency sweep of 500 MHz and a WURST amplitude profile with $n = 16$. The simulations also included different half gaussian $B_1$-distributions shown in the insert. For more details on the simulations see Sect. 2.2.1. (b). Experimental chirped Hahn echo sequence with the same pulses as in (a) and with a sample which extends beyond the resonator (red) and one which fills the resonator but does not extend beyond (black). The sample contained TEIO-N immobilized in trehalose (for more details see the appendix A2).**

distribution with a value of $\sigma = 0.05$ (Fig. 3 (a), green) was used for further simulations, since it closely resembles the
experimental curve with the large sample (Fig. 3 (b), black). We also observed experimentally that shorter pulses seem to
perform better in 2-pulse chirp echo experiments (data not shown). Figure 4 shows a Bloch vector simulation for chirped
2-pulse echoes with different pulse lengths and a fixed half Gaussian $B_1$ distribution with $\sigma = 0.05$. One can observe that both
the maximum echo intensity and the value of $Q_{crit}$ at the maximum is higher for shorter pulses. This is explained by the
different slopes of $\phi_0(Q_{crit})$ for different $TBP$s that we discussed in the previous section. For higher values of $Q_{crit}$ the
difference in echo intensities becomes even more pronounced. Therefore, it is important to choose pulse parameters that give
as short pulses as possible, while still observing a maximum in the echo intensity as a function of $Q_{crit}$. Since this destructive
interference stems from different dynamic phase shifts, the effect is much less pronounced when an even number of $\pi$-pulses
is used to refocus the magnetization. In a sequence with an even number of $\pi$-pulses, $\Delta\phi$ will be refocused completely and the
slope of $\phi_0(Q_{crit})$ will be much flatter, as discussed in the previous section. To illustrate this, Fig. 4 shows different chirped
refocused echo sequences with an increasing number of refocusing $\pi$-pulses. All simulations were performed with the same





half Gaussian $B_1$ distribution that was used in Fig. 3. One can observe in Fig. 4 that the rise of the echo signal intensity with

increasing $Q_{crit}$ is slower with increasing number of $\pi$-pulses. This is because additional refocusing pulses require an overall

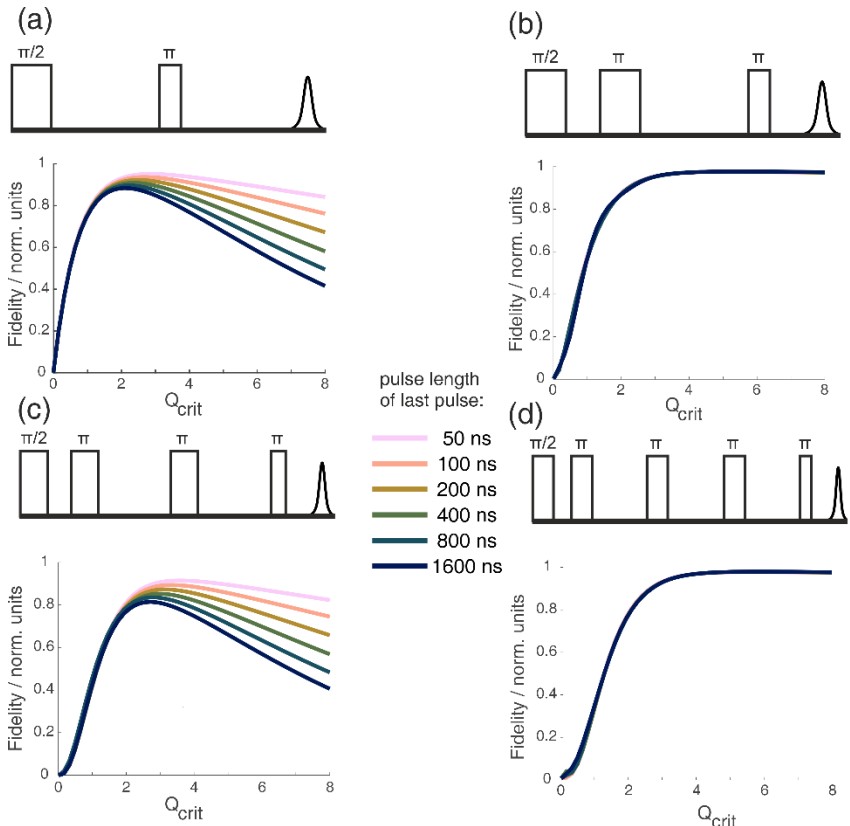

**Figure 4: Bloch vector simulations for different WURST echo or refocused echo sequences with 1, 2, 3 or 4 $\pi$-pulses and different combined pulse lengths of the last two pulses. A half Gaussian B₁ distribution with $\sigma = 0.05$ as shown in Fig. 3 was included in all simulations. The top of each subfigure always shows the pulse sequence used in the simulation below. The simulations show the absolute echo intensity as function of critical adiabaticity $Q_{crit}$ for different pulse lengths of the pulse, while keeping a 2 to 1 ratio between all other pulses and the last pulse. This was done to ensure refocusing of the parabolic phase shift $\phi_p$ as shown in Eq. (8). From (a) to (d) the number of $\pi$-pulses increase.**

higher inversion efficiency, since the coherence needs to be flipped multiple times. Figure 4 ((b) and (d)) clearly show that it

is best to use chirped sequences with an even number of refocusing pulses, if possible. If a sequence with an odd number of

chirped $\pi$-pulses is used, one should aim to minimize the influence of $\phi_0$ and $\Delta\phi$. As is evident from the previous section, it

is best to choose a relatively low $TPB$ to minimize the influence of $\phi_0$ and a relatively high $SW$ to minimize the influence of

$\Delta\phi$. Therefore, short pulses are optimal. In cases with high $B_1$-inhomogeneity, it is especially important to choose pulse

sequences with an even number of $\pi$-pulses and one should choose sequences where the lengths of the $\pi$-pulses are similar,

so that they have a similar $TBP$, and a flat slope of $\phi_0(Q_{crit})$ can be achieved.





## 3 Optimization procedures

### 3.1 Chirped SIFTER pulse calibration

Before setting up echo-FT-EPR sequences with chirped pulses, it is important to verify whether the spectrometer is capable of broadband excitation and detection. Extensive discussions of such calibration experiments have already been published (Doll et al., 2013; Endeward et al., 2023; Spindler et al., 2012). Further information is given in the appendix in Sect. A4. For completeness, we briefly summarize the most significant steps here and supplement them with new considerations.

It is important to check if all amplifiers, in particular the TWT (traveling-wave tube) high-power microwave amplifier, are used in their linear gain regime. It also needs to be verified that the output of the AWG (arbitrary waveform generator) is linear with respect to the input amplitudes for fast amplitude changes (see Sect. A4 and Fig. A2 of the appendix). We have tested this with two different spectrometers, a commercial Bruker spectrometer and our home build X-band spectrometer (Bretschneider et al., 2020), with a different type of AWG, and found that amplitude changes at the fastest time resolution of both AWGs were limited far below the maximum output amplitude of the AWGs, defined by their dynamic range (bit resolution). We adjusted the pre-amplifiers accordingly to make sure that with the reduced AWG amplitudes the TWT amplifier could still be driven to the maximum output intensity (in the linear regime) to achieve maximal $B_1$-field strength with linear amplitude modulations (see Sect. A4 in the appendix for more details).

In the receiver part of the spectrometer, the video amplifier must have the necessary bandwidth and the sampling rate of the transient recorder should be high enough to accurately digitize the broadband signal. In our home-built setup this was not the case, so we recorded the transient signals without the video amplifier using an oscilloscope with a high time resolution and broad enough detection bandwidth. In both spectrometers, there was no clock synchronization of the spectrometer pulses and transient recorder to the AWG. In the case of our home-built spectrometer, since the oscilloscope has a much higher time resolution (0.1 ns compared to 0.625 ns of AWG), this was not a concern.

Setting up a SIFTER sequence for echo-FT-detection needs some additional calibration experiments. We always recorded the resonator profile via a Rabi-nutation experiment, to be able to correct the pulses with the transfer function using the procedure described by Doll et. al (2013). This procedure has also been implemented into the MATLAB package EasySpin (Stoll and Schweiger, 2006). We also recorded nutation experiments, where, in addition to the pulse length, the AWG input amplitude was varied, to get a direct relation between AWG input amplitude and nutation frequency $\omega_1$. This experiment also helps to verify if all amplifiers are used in their linear regime. However, this only controls the linearity at the center frequency $\omega_c$. To determine the linearity of the AWG output over the whole bandwidth other experiments have to be performed, as outlined in Sect. A4.



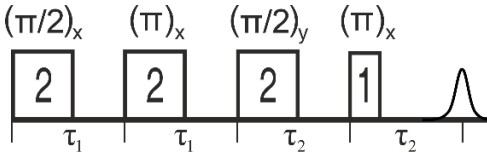

**Figure 5: Schematic representation of the standard 2D-SIFTER sequence with chirped pulses. The pulse length ratios are shown inside the pulses, the pulse delays below the time axis and the pulse flip angles and phases above the pulses.**

Figure 5 shows the 2D-SIFTER sequence, which was first introduced by Doll and Jeschke (2016). Note that the previously introduced sequence used by Schöps et al. (2015), with a pulse length ratio of 2-1-2-1 does not refocus the parabolic phase shift $\phi_p$ and is, therefore, better suited for SIFTER experiments where the echo is integrated and less suited for 2D-SIFTER. The sequence shown in Fig. 5 consists of two $\pi/2$-pulses that have a respective phase shift of 90 °, forming a solid echo, and of two $\pi$-pulses that refocus the in-phase coherence during the application of the second $\pi/2$-pulse and during the echo. The

SIFTER signal is a function of the difference between the pulse delays $\tau_1$ and $\tau_2$ (Jeschke et al., 2000), with its maximum at the position where $\tau_1 = \tau_2$. Therefore, it is best to use equal values of $\tau_1$ and $\tau_2$ when calibrating the pulse amplitudes.

To set up the 2D-SIFTER sequences, we assessed through simulations the shortest possible pulse length of the fourth pulse that would still yield a sufficiently high $Q_{crit}$ with the highest achievable nutation frequency $\omega_1$. The fourth pulse is the $\pi$-pulse with the shortest ratio of the sequence and, therefore, requires the highest nutation frequency $\omega_1$. By determining its length,

the other pulse lengths were fixed through the given ratio of 2-2-2-1. Then we calculated the necessary nutation frequency of the $\pi/2$-pulses with Eq. (7), set up a SIFTER pulse sequence with a fixed amplitude for the $\pi/2$-pulses and swept the amplitudes of the two $\pi$-pulses at the same time with an amplitude ratio of 1 to $\sqrt{2}$ (first to second $\pi$-pulse). We kept $Q_{crit}$ between the two $\pi$-pulses equal to maximize the compensation of their dynamic phase shifts $\Delta\phi$. For a SIFTER sequence with a pulse length ratio 2-2-2-1 this results in the before mentioned amplitude relation of the first and second $\pi$-pulse (2. and 4.

pulse) of 1 to $\sqrt{2}$.





When recording the absolute echo intensity as a function of the $\pi$-pulse amplitudes, we noticed strong oscillations in the echo intensity (Fig. 6 (a) and (b)). This can be explained when considering the phase shift $\phi_0$ in dependence of $Q_{crit}$ as discussed in Sect. 2.2.4. For the SIFTER experiment, the echo that forms at the time of the $(\pi/2)_y$-pulse needs to form along the x-direction so that the pulse does not influence the evolution of the in-phase coherence and only refocuses the dipolar coupling.

Because the echo phase $\phi_0$ is highly dependent on $Q_{crit}$, this condition is only met at specific values of $Q_{crit}$, causing an oscillation in the echo amplitude as a function of the amplitude of the applied $\pi$-pulses. The resulting SIFTER echo intensity is maximal if $\phi_0$ at the $(\pi/2)_y$-pulse is equal to half-integer multiples of $\pi$ and minimal when $\phi_0$ is equal to a multiple of $\pi$. We have simulated the SIFTER echo with the Bloch vector model considering isolated spins without dipolar coupling. Such a simplified calculation already reproduced the experimentally observed amplitude oscillations (see Fig. 6 (b)). The difference

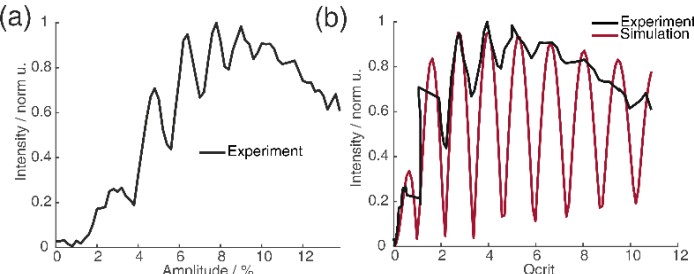

Figure 6: (a) Absolute echo intensity of 4 pulse-SIFTER echo at $\tau_1 = \tau_2$ as a function of AWG amplitude. (b) Intensity of 4 pulse-SIFTER echo as a function of $Q_{crit}$ in black (values calculated from (a)) and Bloch vector simulation using the same pulse sequence in red.

in the amplitude of the oscillations between experiment and simulation might be due to $B_1$-inhomogeneity in the experiment. Recording the SIFTER echo amplitude as a function of the $\pi$-pulse intensity allows to choose the maximum value of $Q_{crit}$, where the absolute echo intensity is high. This will lead to the optimal excitation efficiency and, therefore, also largest dipolar modulation of the SIFTER time trace. After setting the $\pi$-pulse amplitudes to this optimal value, we reevaluated if changing the amplitude of the $\pi/2$-pulses lead to any noticeable improvement in the echo intensity. In most cases, we did not observe

an improvement. The described procedure was more convenient than optimizing the amplitudes of all pulses of the SIFTER sequence individually.

Since the SIFTER experiment is symmetric around $\tau_1 = \tau_2$, one can, in principle, choose whether to perform the SIFTER experiment with values of $\tau_1 \geq \tau_2$, $\tau_1 \leq \tau_2$ or to record both sides symmetrically. Although all methods are viable, we have illustrated in Fig. 7, to avoid echo crossings, overlapping FIDs and the overlap of the left shoulder of the echo with the

protection gate, we recommend to choose to increase $\tau_2$ and decrease $\tau_1$ ($\tau_1 \leq \tau_2$) during the experiment (right side in Fig. 7). Using this approach, the first and second pulse lengths will determine the shortest possible evolution time $\tau_1$. For a chosen maximum echo time $\tau_1 + \tau_2$ (limited by the T$_2$ relaxation time) this defines the highest accessible dipolar evolution time $\tau_2 - \tau_1$. This is obvious by looking at the dipolar evolution blocks shown in Fig. 7. The dipolar evolution time in the SIFTER experiment is the difference of these two blocks (orange and blue). If we choose to increase $\tau_2$ and decrease $\tau_1$ ($\tau_1 \leq \tau_2$) the



maximal evolution time will occur when $\tau_1$ is minimal for which the first, second and third pulse need to be brought as close as possible to each other. In principle, one can then choose a sequence in which the pulse lengths of the first and second pulse are short. Since the pulse lengths of typical chirped pulses can be several 100 ns long, this is of relevance, in particular for short $T_2$ times. In Sect. 4.2.1 we discuss different pulse length combinations that could be used to achieve shorter SIFTER pulse sequences.


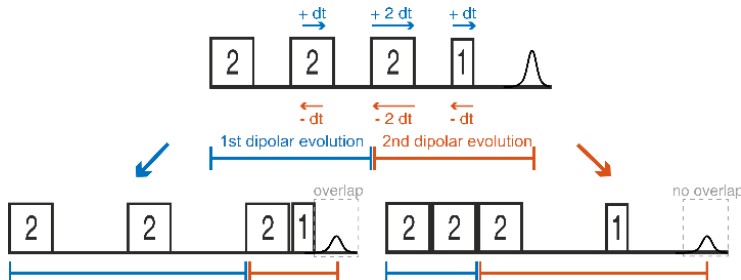

**Figure 7: Two possible sweep directions for the SIFTER experiment. The sweep direction scheme shown on the right side might lead to overlap of the ringing after the last pulse with the left shoulder of the echo signal and can lead to the crossing of other unwanted echoes if the phase cycle is not perfect.**

## 3.2 Comparison of SIFTER echo FT and echo-detected field sweep

We recorded echo-detected field sweep (EDFS) spectra with long monochromatic rectangular pulses of 32 ns and 64 ns to quantitatively compare them with the spectra obtained by Fourier transformation (FT) of the SIFTER echo. The pulses used for recording the SIFTER echo were corrected with the transfer function obtained from the resonator profile. In Fig. 8 (b) we

compare the EDFS with the FT of the SIFTER echo (with pulses before and after correction by the resonator profile Fig. 8 (a)). It can be seen that correcting the pulses with the transfer function obtained from the resonator profile improves the shape of the spectrum obtained by FT of the SIFTER echo. Even with the corrected pulses, we still observe a small deviation from the EDFS spectrum, more pronounced at the high frequency edge of the nitroxide spectrum. The reason for this could be that the transfer function that was obtained from the resonator profile is not perfect or that standing waves also lead to distortions in

the microwave pathway between resonator and detection mixer (a more detailed discussion of this can be found in the literature (Endeward et al., 2023)).





We have found a simple and reliable method to correct for this remaining discrepancy by recording multiple SIFTER echoes while changing the magnetic field. This moves the spectrum into different regions of the excitation bandwidth of the shaped pulses. If both the excitation of the spins and the detection of the signal would be uniform over the full bandwidth, the nitroxide

spectral shape should not change for shifts inside this bandwidth. By choosing and following any specific point in the spectrum and observing how its intensity changes with the magnetic field a profile is obtained that includes both imperfections in

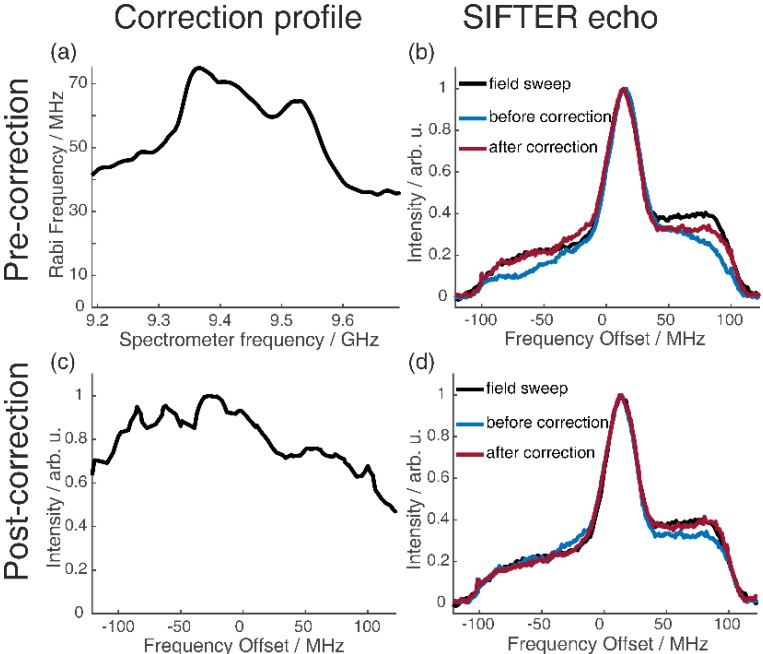

**Figure 8: Correction procedure to get accurate agreement between the FT of the SIFTER echo and the echo detected field sweep. The upper part shows the pre-recording correction of the pulses and the lower part the post-recording correction of the recorded SIFTER echo FT. (a) Resonator profile in the range of the pulse sweep width. (b) EDFS in black, FT of the SIFTER echo without pulse correction (blue) and with pulse correction (red). (c) SIFTER echo excitation profile (see main text and appendix A5 for details). (d) EDFS in black, FT of SIFTER echo with corrected pulses in blue, FT of SIFTER echo divided by excitation profile in (c) in red.**

excitation and refocusing of spins as well as in the detection of the signal. Dividing the FT of the SIFTER echo by this normalized profile results in the red trace in Fig. 8 (d), which matches the EDFS perfectly. In principle, this post-correction method can also be used with uncorrected pulses, but if there is not sufficient excitation at some part of the spectra this will

increase the noise at that offset. It should be noted that this post-correction does not replace optimized excitation in the SIFTER experiment by including the transfer function; it simply reproduces the correct spectral shape. This can be useful if the spectral shape is unknown or if it contains any information that is important for the experiment. For the SIFTER experiments shown in Sect. 4, this post-correction was not performed, because the spectral shape of the SIFTER echo FT already matched the EDFS of the nitroxide well and since the exact spectral shape was not critical for these particular experiments. Importantly,

the great agreement between the SIFTER echo FT and the EDFS was achieved only after identifying, through initial testing, a spectrometer carrier frequency at which standing waves did not significantly distort the microwave pulse shape or the detection.





However, for other experimental setups or spectrometer configurations, where the influence of standing waves is more pronounced, applying such a post-correction might be necessary.

## 4    Results and Discussion

Following the discussion of chirped echoes in Sect. 2 and the outline of our optimization procedures in Sect. 3, we first compare in Sect. 4.1 our experimental 2D-SIFTER results with previously published orientation-selective PELDOR data on the same RNA duplex construct (20 base-pairs labelled with two rigid **Çm** spin labels, for more details see Sect. A2 of the appendix). Afterwards, in Sect. 4.2, we show our first experiments with two novel chirped SIFTER sequences that could be of interest for measuring shorter and longer distances.

**4.1 Comparison of 2D-SIFTER and orientation-selective PELDOR**

Usually, PELDOR experiments are performed to accurately determine the distance distribution between two spin labels (Schiemann et al., 2021). For rigid spin labels, the dipolar coupling does not only depend on the distance between the two spins, PDS experiments also encode information on the Euler angles describing the mutual orientation between the two spins (Marko et al., 2009; Prisner et al., 2015). In PELDOR experiments, this can lead to incorrect distance information if the excited

orientations in a PELDOR experiment only represent part of the dipolar Pake-pattern (Jeschke, 2012). Commonly, PELDOR is performed with flexible spin labels to avoid this orientation selection. This can be also achieved by choosing spectral positions for the pump and excitation pulse that include almost all orientations (for example at Q-band frequencies). The flexible labels have the disadvantage that they artificially broaden the distance distributions and, therefore, decrease the accuracy of the distance determination. For orientation-selective PELDOR, rigid spin labels are deliberately used so that the

spin label orientation is correlated to the conformation of the macromolecule and, therefore, also encodes orientation information of the molecule under study. Typically, multiple time traces are recorded in orientation-selective PELDOR with multiple frequency differences between pump- and probe-pulses or at different magnetic field positions with a fixed difference between pump- and probe-pulses (Gauger et al., 2024; Marko et al., 2011). At X-Band frequencies, the pump-pulse is usually applied at the maximum of the spectrum, were almost no orientation selection is present and the spectral position of the

detection pulses is varied. This gives rise to a set of PELDOR traces where the orientation selection is dominated by the detection pulse position. In this case, the PELDOR traces in good approximation exhibit a "single selection" behavior and allow for direct comparison to the 2D-SIFTER experiment, where the traces are always single selection traces (given by the FT of the SIFTER echo signal). The advantage in the SIFTER experiment is that the orientation-dependent information is encoded in the direct time domain and recorded in parallel in a single 2D-experiment rather in a series of experiments as in the

case of orientation-selective PELDOR. This does not only save measurement time, but allows the determination of orientation information over the entire spectrum with a better spectral resolution (determined by the homogeneous linewidth $2/T_2$).




Averaging over the full spectrum (or echo signal) results in a single SIFTER trace without orientation selection which can then be used to extract the distance distribution using conventional methods of PDS analysis.

For the analysis of the SIFTER data, a second experiment was recorded where the third pulse of the sequence was left out,
which can be used to partially correct the SIFTER background signal (Bowen et al., 2018; Spindler et al., 2016). This sequence has been named SIDRE (SIFTER delay refocused echo) by Vanas et al. (2023). We divided our SIFTER traces by the 3 pulse-SIDRE trace and divided this trace by an exponential background. After Tikhonov regularization this procedure resulted in a distance distribution that agrees very well with the distance distribution for this sample obtained from a PELDOR experiment (data taken from Gauger et al. (2024)) recorded at Q-band frequencies (see Fig. 10 (c)).

Figure 9 (a) shows the X-band PELDOR time traces of a doubly **Çm**-labelled RNA duplex taken from Gauger et al. (2024) and slices of the 2D-SIFTER experiment which we performed on the same sample with broadband shaped pulses (for more information on the sample see the appendix in Sect. A2). The pump pulse in the PELDOR experiments was applied on the

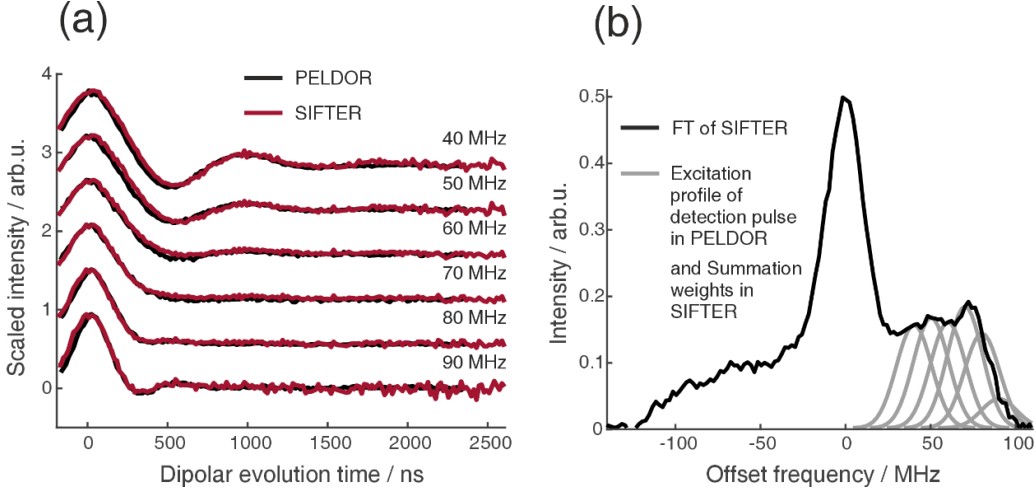

**Figure 9: Comparison of 2D-SIFTER and orientation selective PELDOR time traces of a doubly Çm-labelled RNA duplex recorded at X-band frequencies. (a) Orientation selective PELDOR traces with detection pulses applied at offsets $40 - 90$ MHz from the position of the pump pulse (black) and slices of the 2D-SIFTER experiment weighted by the Gaussian-shaped pulse excitation profiles of the detection pulses used in the PELDOR experiment (red). The SIFTER traces were first divided by the corresponding SIDRE traces and afterwards by an exponential; for the PELDOR traces a typical background correction was applied. Afterwards, all traces were scaled to the same modulation depth. (b) FT of the SIFTER echo at $\tau_2 - \tau_1 = 0$ in black and excitation profiles of the detection pulses used in the PELDOR experiments shown in light grey for the different offset positions. The excitation profiles correspond to the summation weighting for generation of the SIFTER slices shown in (a).**

main peak of the X-Band nitroxide spectrum and the detection pulses were positioned at different offset frequencies from the pump pulse. Figure 9 (b) shows the excitation profile of the Gaussian-shaped detection pulses used in the PELDOR
experiment. The profiles were used for the weighting in the summation of the 2D-SIFTER slices. The SIFTER signal has, as expected, considerably larger modulation depth compared to the PELDOR experiment. For easier visual comparison we have scaled the PELDOR formfactors to the same modulation depth.

The experimental datasets obtained by orientation-selective PELDOR and 2D-SIFTER match very well and the traces have similar signal-to-noise ratio (SNR) even though the PELDOR traces took 5 days to record, with new optimizations for every

offset, and the SIFTER experiment including the background signal experiment took only 24 hours. In principle, this acquisition time could be further decreased since the theoretical acquisition time with the used SRT and number of averages in the SIFTER experiment was only 12 hours. Since in our home-built setup we reprogrammed the AWG for every step including 8 different tau-averaging steps, we recorded the data with an idle time of about 50 %.

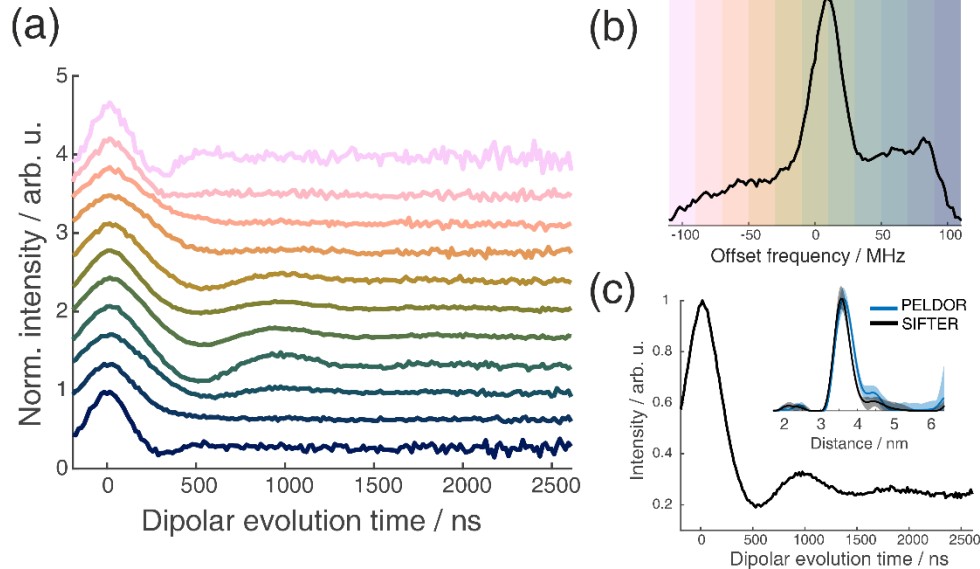

**Figure 10: Full 2D-SIFTER data set of a doubly Çm-labelled RNA duplex (a) Slices of 2D-SIFTER experiment summed over areas shaded in (b). (b) FT of SIFTER echo at $\tau_2 - \tau_1 = 0$ and frequency slices of SIFTER data in (a). Here the SIFTER traces are only divided by their corresponding SIDRE traces and no additional exponential decay was subtracted. (c) 2D-SIFTER slice over whole spectral range shown in (b) the inlet shows the distance distribution obtained from (c) (black) and the distance distribution of a PELDOR experiment at Q-band (blue) where no orientation selection should be present.**

In orientation-selective PELDOR, the frequency resolution is limited by the excitation profile of the pulses and by the overlap

between excitation and pump pulses. SIFTER does not have these limitations and the frequency resolution is only limited by the homogenous linewidth and the SNR of the recorded data. This is shown in Fig. 10 (a) where all traces at the positions marked in Fig. 10 (b) over the X-Band spectrum are shown. Here vertical slices are added together to achieve sufficient SNR. The data show very pronounced orientation selectivity; one can clearly observe the dispersion in dipolar frequencies ranging from $\omega_{dd}$ in the center to $2\,\omega_{dd}$ at the edges of the spectrum. Alternatively, one can also average over all orientations and

obtain a SIFTER trace with no orientation selection as shown in Fig. 10 (c) which gives the correct distance distribution. Because of the rigid label used, this distance distribution is more accurate than one obtained with PELDOR spectroscopy and





a flexible label, while the data still also contains all the orientation information. To obtain an accurate distance distribution with rigid labels and PELDOR spectroscopy one has to record the data set at Q-band where the spectral dispersion caused by g- and A-anisotropies are of similar magnitude, causing a strong spectral overlap of different orientations (Gauger et al., 2024).

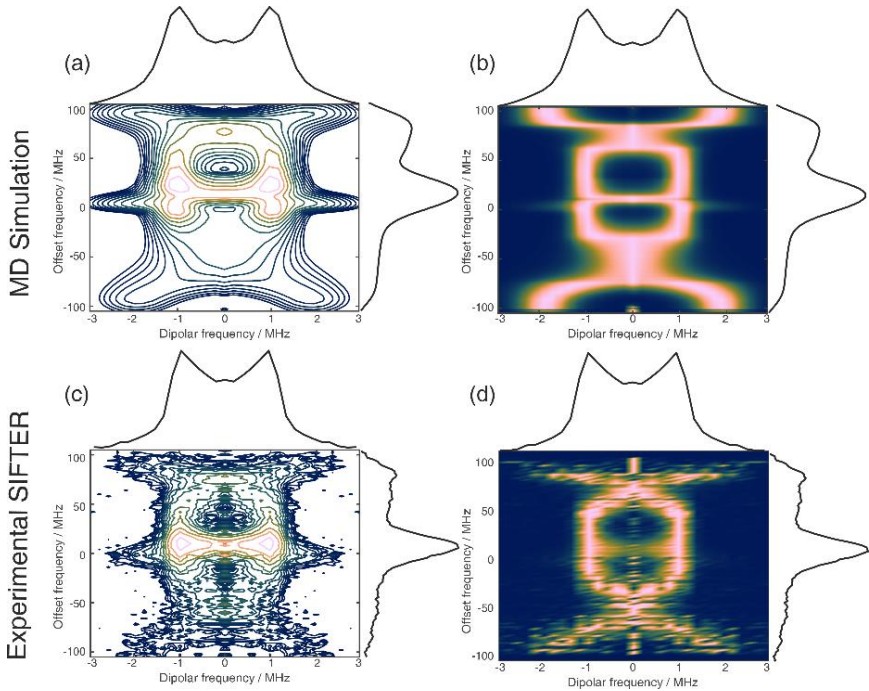

**Figure 11: 2D FT of SIFTER data yielding a 2D EPR-correlated SIFTER spectrum. (a) and (b) show the data simulated from conformers obtained from MD simulations with explicitly modelled spin labels taken from Gauger et al. (2024) and (c) and (d) show the experimental data of the 4 pulse-SIFTER experiment with a pulse length ratio of 2-2-2-1. The dipolar spectrum shown above the correlation plots is the sum of all dipolar spectra and the EPR spectrum shown on the side is the real part of the direct FT at the maximum of the time trace. (a) and (c) are contour plots with logarithmically spaced contour lines of the 2D FT of the simulated form factors and the experimental SIFTER echoes respectively. (b) and (d) show a heatmap of the 2D FT data sets where each dipolar spectrum was normalized and squared to facilitate visually following the main features of the dipolar spectrum.**

Instead of adding vertical slices together, as was shown in Fig. 10 (a), it is also possible two obtain a 2D dipolar spectrum which is correlated by the EPR spectrum of the nitroxide (Fig. 11 (c)). This is achieved by performing an additional FT along the dipolar time axis. After the first FT along the EPR dimension, we applied the previously described background correction procedure to each SIFTER time trace. Each trace was then normalized and multiplied by its intensity prior to background correction. Finally, we applied a magnitude FT without any zero filling or apodization along the dipolar time axis. The resulting

2D EPR-correlated SIFTER data are visualized in Fig. 11, where we compare the experimentally obtained data ((c) and (d)) with a spectrum simulated from conformers obtained by a molecular dynamics (MD) simulation ((a) and (b)). The conformers were taken from Gauger et al. (2024), where the authors simulated the RNA duplex with explicitly included spin labels and verified the MD simulations with a full orientation-selective PELDOR data set recorded at X-, Q- and G-band. Here we have simulated time traces for each offset frequency from this conformer set, by the same procedure that was previously used for





orientation-selective PELDOR (Marko et al., 2010; Stelzl et al., 2017) and that was adapted for SIFTER by Bowen et al. (2018). In Fig. 11 (a) and (c) we show a contour plot with logarithmically spaced contour lines that encodes both intensities of the EPR-spectrum and the dipolar spectrum. In Fig. 11 (b) and (d) we have normalized each dipolar spectrum to facilitate visual inspection of the main features of the dipolar spectrum. Towards the edges of the EPR spectrum the bad SNR made it difficult to properly fit an exponential to the data and a zero-frequency artefact is still present in the 2D data set. Nevertheless, experiment and simulation match very well, with the slight deviations being expected as they were also present between the orientation-selective PELDOR data set and the MD simulations (Gauger et al., 2024). Overall, it is clear that 2D-EPR correlated SIFTER data can be a great addition to orientation-selective dipolar studies.

## 4.2 First experiments with chirp-SIFTER variations

### 4.2.1 Different pulse length ratios in 4 pulse-SIFTER

As explained in Seq. 2.2 one can in principle use Eq. (8) to find other pulse length and sweep width combinations that result in a refocusing of the parabolic phase shift $\phi_p$, similar as the Böhlen-Bodenhausen scheme for a two-pulse echo (Bohlen et al., 1989). In this section we discuss such solutions of Eq. (8) for the SIFTER sequence.

The rational for the 2-2-2-1 SIFTER scheme used by Doll and Jeschke (2016) was that the third pulse should not affect the in-phase coherence terms, which implies that the echo produced by the first and second pulse should have the same frequency dispersion as the third pulse. As we have shown in Sect. 3.1 this also requires that the phase $\phi_0$ of the first two pulses actually results in a phase along the y-direction. One of the major drawbacks of the 2-2-2-1 sequence is that the dipolar coupling evolution times are actually not free from offset frequency dispersion (Doll, 2016; Doll and Jeschke, 2016), which makes the sequence not suitable for short distances. When considering the dipolar coupling, to first approximation, only the solid echo of the first and third pulse needs to be considered (if we ignore partial refocusing during the chirped pulses (Doll, 2016)). The 2-2 ratio between the two $\pi/2$-pulses in the 2-2-2-1 SIFTER sequence is not able to refocus the dipolar coupling frequencies for all offset frequencies at one point in time but rather does so with an offset frequency dispersion. We can also solve this computationally by adding to Eq. (8) a coherence pathway with coherence order +1 before the third pulse and -1 afterwards. If we require that the offset frequency dispersion, introduced by pulses 1 and 2, be the same as the offset frequency dispersion of the third pulse and additionally require that pulses 1 and 3 refocus the dipolar coupling without offset frequency dispersion, then the pulse length ratio 4-3-2-1 with only up sweeps (i.e. from low to high frequencies) is a unique solution. However, this sequence has very unfavourable relative pulse lengths (see discussion of shortest $\pi$-pulse in a sequence in Sect. 3.1) and was not tested in this work.

We have investigated pulse sequences where the offset frequency dispersion created by the first two pulses is explicitly not equal to the frequency dispersion of the third pulse. In this case, the third pulse does affect the evolution of the in-phase coherence of all the spins that are orientated along the x-axis at the time of passage during the third pulse by introducing an additional coherence transfer. This leads to a second coherence pathway as is the case when using monochromatic pulses





(Borbat and Freed, 2017). This can be investigated particularly well by a SIFTER pulse sequence with a pulse length ratio of 2-2-6-3, with sweep directions up-down-down-down. This sequence refocuses the frequency offsets for both coherence pathways of the SIFTER sequence, but does so at different times. While this sequence performs worse for SIFTER experiments, since it introduces a large offset frequency dispersion in the refocusing time of the dipolar coupling, it allows the study of the two different coherence pathways separately. In Figure 12 (a) and (b) the refocusing times of the two different echoes are shown in simulation and experiment. In Fig. 12 (c) we compare the echo phase $\phi_0(Q_{crit})$ of experiment and simulation, which agree very well. Echo 1 corresponds to the coherence pathway where the third pulse causes coherence inversion and echo 2 corresponds to the regular refocused echo coherence pathway. Because the pathway of echo 1 includes an uneven number of coherence inversions, the phase shift $\phi_0$ is not refocused, leading to a strong dependence of $\phi_0$ on $Q_{crit}$. Our simple Bloch vector model simulation does not reproduce the difference in echo intensity between the two echoes (Fig. 12 (b)). However, we have observed a rather large decay of the echo intensity with higher values of $Q_{crit}$ (Fig. C11), which indicates that the distribution of $\phi_0$ with a large $B_1$-inhomogeneity might explain this difference. Figure 12 (d) shows the echo intensity of the two echoes during a SIFTER experiment. Both echo signals show the expected dipolar oscillation frequency





as a function of the evolution time and result in similar distance distributions, compared to the PELDOR experiment (see appendix C3 Fig. C12 and C13).

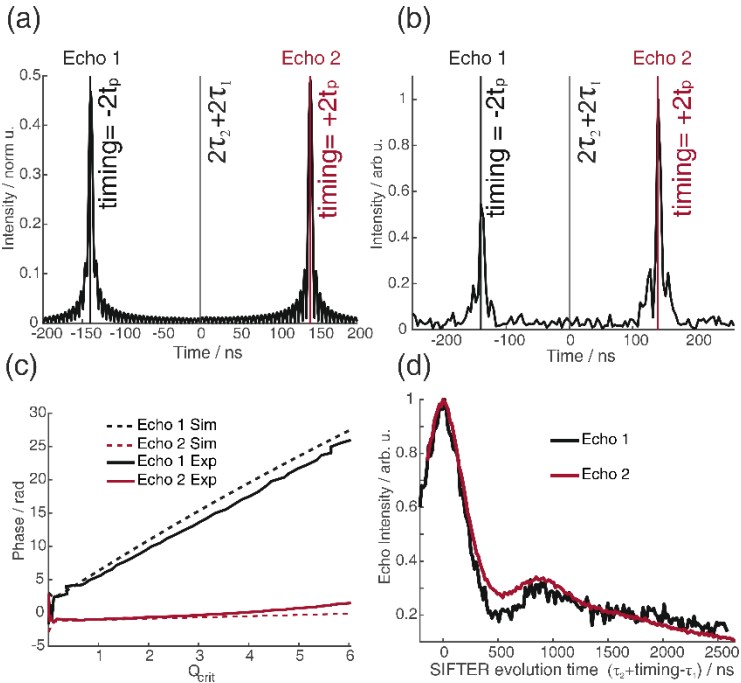

**Figure 12: Bloch vector simulations (a) and experimental results of echo transient (b) of the 4 pulse-SIFTER sequence with a pulse length ratio of 2-2-6-3 and sweep directions up-down-down-down. Echo 1 corresponds to the coherence pathway where the resonance offset is affected by the $(\pi/2)_y$-pulse and echo 2 corresponds to the regular pathway where the spins are unaffected by the third pulse. (a) The simulation is normalized to the number of spins, i.e. the intensity would be one if all spins are refocused during an echo. (b) The experiment was normalized to the maximum of echo 2. (c) Comparison of the phase of the two echoes in simulation (dashed line) and experiment (solid line). (d) Experimental SIFTER time traces of both echoes (shifted by the timings shown in (b) to overlay). The sample is a doubly Çm-labelled RNA duplex.**

This observation in led us to explore different pulse-length ratios for which the offset-frequency dispersion requirement of the echo during the third pulse is not fulfilled, but which should have less offset-frequency dispersion of the dipolar refocusing time. Such a sequence could be favourable for broadband SIFTER measurements of short distances.

Therefore, we considered only the regular refocused echo pathway required that the refocusing of dipolar frequencies by the solid echo should be without frequency dispersion. This was done computationally by considering two coherence pathways (one for the refocused echo and one for the solid echo) and solving Eq. (8), which led us to a sequence with a pulse length ratio 2-3-1-4 and the sweep directions up-down-up-down (there are also numerous other similar solutions). The 2 to 1 relationship between the first and the third pulse ensures refocusing of the dipolar coupling without offset frequency dispersion.

The relative pulse lengths are much more favourable than with the 4-3-2-1 sequence, since in the new sequence the relative pulse lengths of the $\pi/2$-pulses are short, compared to the pulse lengths of the $\pi$-pulses. This sequence will give only half the signal intensity since only magnetization which arrives with y-phase during the $(\pi/2)_y$-pulse will be refocused. Nevertheless, the sequence might be of interest for shorter distances were the 2-2-2-1 SIFTER sequence is not so well-suited. We recorded



a SIFTER trace with this new sequence and obtained a good SNR (see Fig. 13 (b)). To verify that the sequence is actually

suitable to record distance distributions with short distances, experiments with samples with distances shorter than 2.5 nm should be performed. Also a proper analysis by operator formalism as performed for the 2-2-2-1 sequence (Doll and Jeschke, 2016) would be necessary, to make sure that partial refocusing during the pulses does not make this sequence infeasible for short distances.

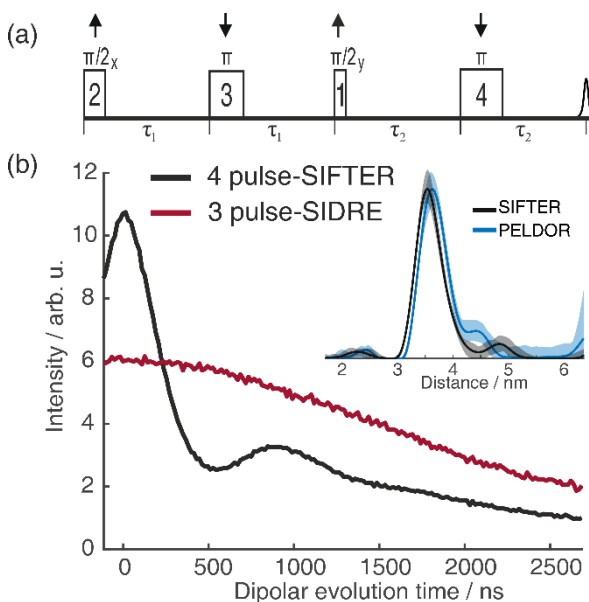

**Figure 13: 4 pulse-SIFTER sequence with a pulse length ratio of 2-3-1-4. (a) Schematic representation of the sequence including the pulse length ratio, with sweep directions indicated by arrows above the pulses and the pulse delays below the time axis. (b) 4 pulse-SIFTER and 3 pulse-SIDRE trace averaged over the entire EPR-spectrum to remove orientation selection. The 3 pulse-SIDRE trace was recorded with roughly have the averages compared to the 4 pulse-SIFTER trace (for more details see Sect. A3). The insert shows the distance distribution obtained with this SIFTER trace (black) compared to the distance distribution obtained by PELDOR at Q-band. The sample is a doubly Çm-labelled RNA duplex (Details in appendix Sect. A2).**

### 4.2.2 Carr-Purcell 6 pulse-SIFTER

In PELDOR spectroscopy, several Carr-Purcell (CP) type sequence have been introduced that make use of the slower decay of transverse magnetization in CP echo trains (Borbat et al., 2013; Spindler et al., 2013, 2015; Tait and Stoll, 2016). Similar concepts can be realized in SIFTER sequences by introducing additional refocusing pulses. As shown in Fig. 4 it is best to introduce two additional $\pi$-pulses since an even number of $\pi$-pulses can remove the dispersion in $\phi_0$ and $\Delta\phi$ due to $B_1$

inhomogeneity. In principle, one can insert these additional $\pi$-pulses on either side of the $(\pi/2)_y$-pulse, leading to a symmetric





6 pulse-SIFTER experiment, or both on one side of the $(\pi/2)_y$-pulse leading to an asymmetric 6 pulse-SIFTER sequence (Fig. 14 (a) and (b)). We decided to record the asymmetric 6 pulse-SIFTER sequence, because introducing additional $\pi$-pulses on both sides of the $(\pi/2)_y$-pulse would increase the overall length of the sequence, since one cannot have overlapping pulses. In Fig. 14 we show the pulse positions with maximum and minimum dipolar evolution time. An additional pulse on the left

side of the $(\pi/2)_y$-pulse in Fig. 14 (b) would increase the length of the first dipolar evolution block while decreasing the length of the second and, therefore, decreasing the maximum dipolar evolution time.

The asymmetric sequence should also have a more advantageous timing of the optimal dynamic decoupling position. The optimal refocusing occurs when all $\pi$-pulses are spaced out evenly, or in other words, when all inter-pulse delays $\tau$ are equal. For the asymmetric 6 pulse-SIFTER sequence this will occur later in the time trace, where the signal modulation is damped

further. In the classical 4 pulse-SIFTER experiment the dynamic decoupling is optimal at the position $\tau_1 = \tau_2$, where the

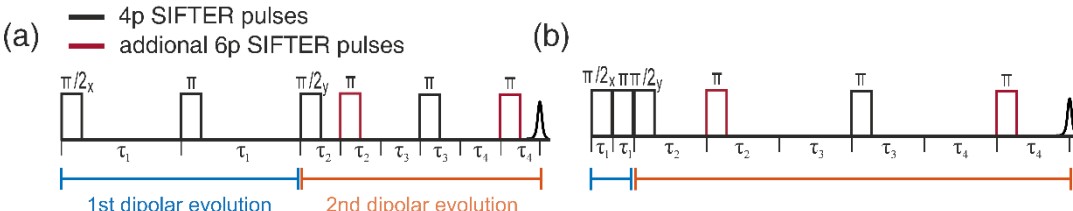

**Figure 14: Schematic representation of the asymmetric 6 pulse-SIFTER sequence. The two additional $\pi$-pulses are shown in red. The first and second dipolar evolution blocks are shown in blue and orange underneath the time axis. (a) Timing of pulses at full dipolar refocusing, (b) during maximal dipolar evolution.**

overall SNR should be best, since the largest signal modulation is expected at this time. One can observe the position of the best dynamic decoupling time by observing when the maximum of the SIDRE trace occurs. In all our 3 pulse-SIDRE experiments this position was observed approximately at $\tau_1 = \tau_2$. Another advantage of the 6 pulse-SIFTER sequence lies in the pulse length ratio. Somewhat counterintuitively the 2-2-2-2-3-2 6 pulse-SIFTER sequence can be made shorter than the

conventional 2-2-2-1 4 pulse-SIFTER experiment with realistic pulse lengths of the broadband pulses. This is the case because the shortest $\pi$-pulse in a sequence determines how short the overall sequence can be made (this was discussed in Sect. 3.1). Similar to the 4 pulse-SIFTER sequence the 6 pulse-SIFTER time trace is still a function of the difference of the two dipolar evolution blocks, but due the additional $\pi$-pulses the echo will be modulated by $S\big((\tau_2 + \tau_3 + \tau_4) - \tau_1\big)$ (pulse delays indicated in Fig. 15 (a)). During the experiment, $\tau_1$ is decreased by $dt$ while $\tau_2$, $\tau_3$ and $\tau_4$ are increased by $dt/3$. This results in the

pulses moving by the increments shown in Fig. 15 (a).

We recorded a 6-pulse-SIFTER trace and the 5-pulse equivalent to the 3-pulse SIDRE (the second $\pi/2$-pulse is omitted). The sequence produces pronounced dipolar oscillations with the expected dipolar frequency and with a similar SNR compared to the 4 pulse-SIFTER experiment with the same number of averages (Fig. 15 (b)). We also observe a much flatter background (Fig. 15 (b)) in the 6 pulse-SIFTER time trace as well as in the 5 pulse-SIDRE trace (Fig. C6 in the appendix). Therefore, the

SNR of the 6 pulse-SIFTER experiment should be better after background correction. To make this sequence applicable for





quantitative distance measurements it would be necessary to thoroughly investigate its background signals, as was done for the 4 pulse-SIFTER experiment by Vanas et al. (2023).

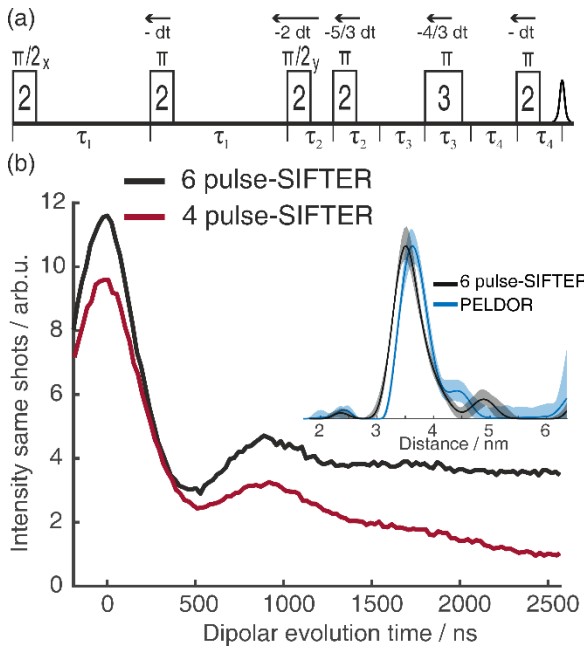

**Figure 15: (a) 6 pulse-SIFTER pulse sequence with pulse delays τ, pulse length ratio and pulse movement during sequence. (b) 6 pulse-SIFTER on a doubly labelled RNA duplex (black) and 4 pulse-SIFTER with the same number of scans (red). Both traces are shown without any background correction. The inlet shows the distance distribution obtained with this 6 pulse-SIFTER trace (black) compared with the distance distribution obtained by PELDOR at Q-band blue.**

## 5    Conclusion

In this study, we present deep insights into the application of chirped pulses for echo-FT-EPR and examine several of the challenges associated with their use in broadband detection of transverse magnetization in the field of in EPR. We provide a detailed discussion on the proper design of chirped pulse sequences to ensure accurate representation of the Fourier spectrum obtained from broadband echo signals. Following this procedure, we demonstrate how to obtain accurate 2D-SIFTER data at X-band frequencies.

In particular, we investigated an RNA duplex construct consisting of 20 base-pairs labelled with two rigid **Çm** spin labels by 2D-SIFTER and compared the experimental results to already published orientation-selective X-band PELDOR data (Gauger et al., 2024). We have shown that at X-band frequencies, where the pump pulse excites almost all orientations in orientation-selective PELDOR, 2D-SIFTER, as an inherently single frequency technique, can yield similar results in a shorter



experimental time while providing more detailed and better resolved orientation information that is encoded in the frequency
domain.

In general, 2D-SIFTER offers the significant advantage that one can obtain the optimal resolved orientation-selective information, or eliminate all orientation selection, by summing over the full echo trace. The use of the rigid **Çm** spin labels is essential to get accurate orientation information and provides distances that are more accurate compared to flexible spin labels. We also introduce a 4 pulse-SIFTER experiment with a different pulse length ratio, which might be of interest when studying
shorter distance distributions where the typical 2-2-2-1 SIFTER sequence fails.

By applying dynamic decoupling approaches that have been used successfully in the past with PELDOR, we introduced a 6 pulse-SIFTER sequence by extending the regular 4 pulse-SIFTER experiment with two additional $\pi$-pulses. The pulse sequence seems to have a reduced intermolecular background contribution, which will have to be investigated further before using such sequences for quantitative distance measurements.

We believe that 2D-SIFTER with shaped pulses and rigid spin labels has great advantages, compared to orientation-selective PELDOR, and that the limits of the technique are still far from being fully explored.



## Appendix A: Details on sample, set-up and acquisition of data in main text

### A1 Hardware

All experiments were recorded on a home-built X-band spectrometer (Bretschneider et al., 2020). The experiments were performed with a fully over-coupled BRUKER ER41118X-MS3 split-ring resonator. The temperature was achieved with a continuous-flow helium cryostat (CF935, Oxford Instruments) and controlled by a temperature control system (ITC 502, Oxford Instruments). The Microwave carrier frequency was generated by a MW synthesizer (HP 8672A). The carrier frequency was amplitude and phase modulated by an arbitrary waveform generator (AWG, SD devices SDR14) and amplified

by a 1 kW traveling-wave tube amplifier (TWT, Applied Systems Engineering 117X). The AWG has a time resolution of 0.625 ns (1.6 Gs/s) and a 14-bit amplitude resolution. The AWG is controlled by MATLAB code and the AWG controls the I- and Q-ports of two microwave mixers. The carrier frequency input to the two mixers is phase shifted by 0 ° and 90 ° respectively and the output of the two mixers is combined to create the pulse shapes. The detection of the EPR signals was either achieved by a Bruker *ElexSys* SpecJet transient recorder (4 ns time resolution, 8-bit amplitude resolution) with a

200 MHz video amplifier or without the video amplifier and with and Rohde & Schwarz RTO 1024 oscilloscope (0.1 ns time resolution (10 Gs/s)).

All SIFTER experiments were carried out at 50 K and were recorded with the oscilloscope and without the video amplifier. While the Hahn echo experiments shown in Fig. 3 were recorded at ambient temperature with the video amplifier and the transient recorder. All SIFTER echo vs $\pi$-pulse amplitude experiments shown in the appendix were also recorded with the

video amplifier and the transient recorder but at 50 K.

### A2 Sample

Two samples were used in this study. All SIFTER measurements were performed on a doubly **Çm**-labelled RNA duplex, while the Hahn experiments shown in Fig. 3 were performed on a 1,1,3,3-tetraethylisoindolin-5-isthiocyanate-2-oxyl (TEIO-N) nitroxide immobilized on trehalose, with an approximate spin concentration of 1 mM (Saha et al., 2015). A more detailed

description of the RNA duplex sample is given by Gauger et al (2024). We provide only the most relevant aspects. The synthesis and purification of the oligonucleotides was performed in the lab of S. Th. Sigurdsson by Dr. Anna-Lena J. Halbritter and the EPR sample preparation was done by Dr. Maximilian Gauger in the lab of T.H. Prisner.

The rigid spin label used in labelling is a cytidine-analogue, which was introduced by Höbartner et al. (2012) and is referred to by the symbol **Çm**. The labels are rigid cytidine-analogues which allow high precision distance measurements and makes

it possible to determine the mutual orientation of the two spin labels.

The position of the label in the RNA duplex is indicated by two numbers in parenthesis. The sample used in this study was RNA(3,17). Here **Çm** was introduced at the third position counting from the 5'-end (the additional uracil overhang, which is introduced to prevent end-to-end stacking is not counted (Erlenbach et al., 2019).) The second label is introduced on the counter strand in position 17 counting from the 5'-end of the first strand.



To prepare the EPR samples purified oligonucleotides were dissolved in an aqueous phosphate buffer (10 mM phosphate, 100 mM NaCl, 0.1 mM Na₂EDTA, pH 7.0) combined with 20 % d6-ethylene glycol and 20 μL and were transferred into 3.2 mm outer diameter Suprasil tubes. The RNA concentration in the sample tubes was 100 μM.

**A3 Experimental details and pulse parameters**

   All shaped pulses used here had a WURST amplitude modulation with a truncation parameter of $n_{wurst} = 16$ and a sweep

width of $SW = 500$ MHz. For all pulses were it is not explicitly mentioned otherwise the sweep direction of the frequency sweep was upward from low to high frequencies (-250 MHz to 250 MHz relative to microwave carrier frequency). The pulses were corrected with the transfer function obtained from the resonator profile by the procedure described by Doll et al. (2013). The resonator profiles are given in Sect. A5 of the appendix.

All 2 pulse-echo experiments were carried out with a simple 2-step phase cycle:

$$\phi_{\pi/2, \ first} = [0°, 180°]$$

$$\phi_{\pi, \ second} = [0°, 0°]$$

$$\phi_{det} = [+, -]$$

   All 4 pulse-SIFTER experiments were carried out with the following 16-step phase cycle and all 3 pulse-SIDRE experiments

were carried out with the same phase cycle but with the third pulse removed:

$$\phi_{\frac{\pi}{2}, \ first} = [0°]_{16}$$

$$\phi_{\pi, \ second} = [0°, 180°]_4, [90°, 270°]_4$$

$$\phi_{\frac{\pi}{2}, \ third} = [90°]_{16}$$

$$\phi_{\pi, \ fourth} = [[0°]_4, [180°]_4]_2$$

$$\phi_{det} = [+]_8, [-]_8$$

   The 6 pulse-SIFTER experiments were carried out with the following 64-step phase cycle and the 5 pulse-SIDRE equivalent were recorded with the same phase cycle but with the third pulse removed:

$$\phi_{\frac{\pi}{2}, \ first} = [0°]_{64}$$

$$\phi_{\pi, \ second} = [[0°]_8, [90°]_8, [180°]_8, [270°]_8]_2$$

$$\phi_{\frac{\pi}{2}, \ third} = [90°]_{64}$$

$$\phi_{\pi, \ fourth} = [[0°]_2, [90°]_2, [180°]_2, [270°]_2]_8$$

$$\phi_{\pi, \ fifth} = [0°, 180°, 90°, 270°]_8, [90°, 270°, 0°, 180°]_8$$

$$\phi_{\pi, \ sixth} = [0°]_{64}$$

$$\phi_{det} = [+]_{32}, [-]_{32}$$





**2-pulse chirp echo**


The 2 pulse-echo experiments with chirped WURST pulses shown in Fig. 3 (b) were recorded with a pulse length parameter $t_p$= 100 ns and a pulse length ratio of 2-1.

**SIFTER with pulse length ratio 2-2-2-1**

The pulse length parameter in the 4 pulse-SIFTER experiment, that is shown in Fig. 9 and Fig. 10 was $t_p = 66.25$ ns with

relative pulse lengths 2-2-2-1. The experiment was carried out with 8 $\tau$-averaging steps of 8 ns length to remove proton ESEEM modulation. The $\tau$-averaging was performed for both $\tau_1$ and $\tau_2$ to not change the SIFTER evolution time. The experiment was carried out with a starting value of $\tau_1 = 1616$ ns and $\tau_2 = 1424$ ns, during the experiment $\tau_1$ was decremented and $\tau_2$ was incremented by $d\tau = 8$ ns in 176 steps. The acquisition parameters were shot repetition time $SRT = 5$ ms, shots per point of $SPP = 1600$ and 2 scans.

For the 4 pulse-SIFTER experiment shown in Fig. 15 the pulse length parameter was $t_p = 210$ ns with relative pulse lengths of 2-2-2-1. Here only the 4 pulse-SIFTER experiment was carried out with the 8 $\tau$-averaging steps of 8 ns length, while the 3 pulse-SIDRE experiment was recorded without $\tau$-averaging. The experiment was carried out with a starting value of $\tau_1 = 1824$ ns and $\tau_2 = 1728$ ns, during the experiment $\tau_1$ was decremented and $\tau_2$ was incremented by $d\tau = 12$ in 116 steps. For the 4 pulse-SIFTER experiment the acquisition parameters were $SRT = 5$ ms, shots per point of $SPP = 2496$ and 2 scans

and for the 3 pulse-SIDRE experiment $SRT = 5$ ms, shots per point of $SPP = 2496$ and 12 scans but without $\tau$-averaging.

**SIFTER with pulse length ratio 2-2-2-2-3-2**

The pulse length parameter in the 6 pulse-SIFTER experiment, that is shown in Fig. 15 was $t_p = 105$ ns with relative pulse lengths 2-2-2-2-3-2. The 6 pulse-SIFTER experiment was carried out with 8 $\tau$-averaging steps of 8 ns for both $\tau_1$ and $\tau_2$, $\tau_3$ and $\tau_4$ were not changed during the $\tau$-averaging. The 5 pulse-SIDRE experiment was not recorded with any $\tau$-averaging. The

starting values were $\tau_1 = 1620$ ns and $\tau_2 = \tau_3 = \tau_4 = 540$ ns, during the experiment $\tau_1$ was decremented with $d\tau = 12$ ns and $\tau_2$, $\tau_3$, and $\tau_4$ were incremented by $d\tau = 4$ ns in 116 steps. The acquisition parameters for the 6 pulse-SIFTER experiment were $SRT = 5$ ms, $SPP = 2496$ and 2 scans and for the 5 pulse-SIDRE experiment $SRT = 5$ ms, $SPP = 1280$ and 12 scans.

**SIFTER with pulse length ratio 2-3-1-4**

The pulse length parameter in the 4 pulse-SIFTER experiment, that is shown in Fig. 13 was $t_p = 70$ ns with relative pulse

lengths 2-3-1-4 and sweep directions up-down-up-down. The experiment was carried out with 8 $\tau$-averaging steps of 8 ns for both $\tau_1$ and $\tau_2$. The starting values were $\tau_1 = 1632$ ns and $\tau_2 = 1448$ ns, during the experiment $\tau_1$ was decremented and $\tau_2$ was incremented by $d\tau = 8$ ns in 176 steps. The acquisition parameters for the 4 pulse-SIFTER experiment were $SRT = 5$ ms, $SPP = 2496$ and 2 scans and for the 3 pulse-SIDRE experiment $SRT = 5$ ms, $SPP = 1280$ and 2 scans.



**SIFTER with pulse length ratio 2-2-6-3**

The pulse length parameter in the amplitude sweep of the $\pi$-pulses of the 4 pulse-SIFTER echo, that is shown in Fig. 12 (a-c), was $t_p = 70$ ns with relative pulse lengths 2-2-6-3 and sweep directions up-down-down-down. The pulse length parameter in the 4 pulse-SIFTER experiment, that is shown in Fig. 12 (d) was $t_p = 33.125$ ns with relative pulse lengths 2-2-6-3 and sweep directions up-down-down-down. The experiment was carried out with 8 $\tau$-averaging steps of 8 ns for both $\tau_1$ and $\tau_2$. The starting values were $\tau_1 = 1492$ ns and $\tau_2 = 1396$ ns, during the experiment $\tau_1$ was decremented and $\tau_2$ was incremented by

$d\tau = 8$ ns in 176 steps. The acquisition parameters for the experiment were $SRT = 5$ ms, $SPP = 1600$ and 2 scans.

**A4 Linearity of amplifiers and AWG**

**Nutation experiments and linearity of AWG**

It is common practice to verify the linearity of the amplifiers in the excitation chain by performing a series of nutation

experiments at varying AWG input amplitudes. These experiments also provide a direct correlation between the AWG input and the resulting nutation frequency $\nu_1$. An example of this is shown in Fig. A1. Ideally, all pre-amplifiers preceding the TWT should be configured to ensure that the nutation frequency increases approximately linearly with the input amplitude. Nevertheless, to maximize the achievable nutation frequencies, the system should be operated near the saturation point of the TWT.

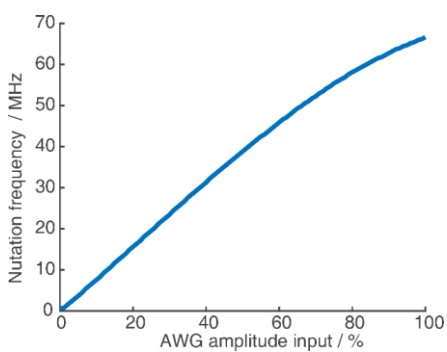

**Figure A 1: Rabi nutation amplitude profile. A Rabi nutation experiment was recorded with different amplitudes of the nutation pulse and the nutation frequency is plotted against the AWG input amplitude. The sample here was a $\gamma$-irradiated quartz sample.**

We observed that in two of our spectrometer setups, the home-built X band system described in Sect. A1 of the appendix and a commercial Bruker system, the AWGs were unable to maintain consistent output amplitudes at higher frequencies. This effect is illustrated in Fig. A2, where in (a) and (b) three long microwave pulses were recorded directly using an oscilloscope, without amplification by the TWT and without the probe head. In each case, the AWG input amplitude was gradually ramped from 0 to 100%. The first pulse (black) had no additional modulation. The second pulse (red) was modulated by inverting the





amplitude every 20 ns, and the third pulse (blue) included the maximal modulation, with the amplitude alternating with the

time resolution of the AWG (0.625 ns). One can see that for the fast oscillations (blue) the AWG is not capable of producing

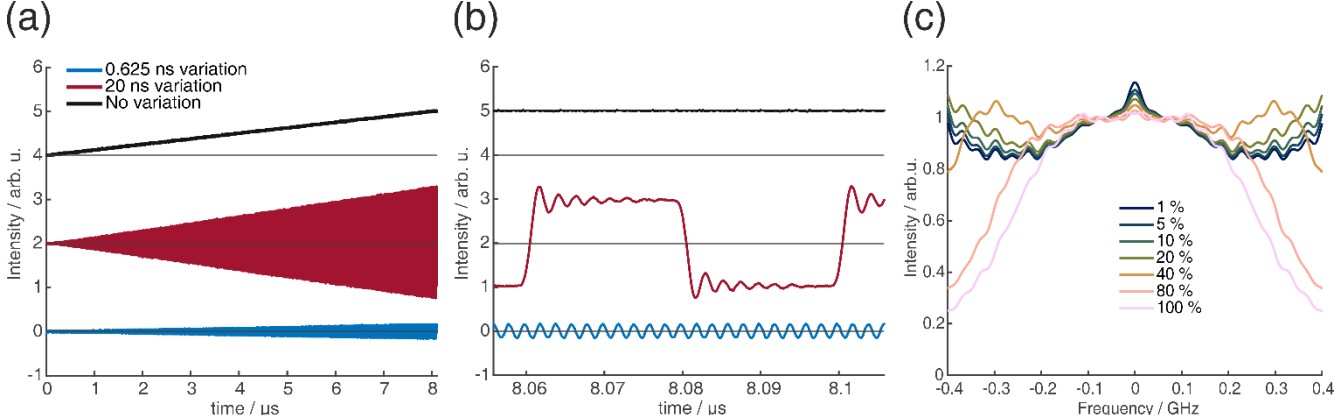

**Figure A 3: AWG speed tests. (a) and (b) Three approximately 8 $\mu$s long microwave pulses with a linearly increased amplitude from 0 to 100 % and additional modulation (No modulation (black), every 20 ns (red) and every 0.625 ns (blue)). (a) The measured output amplitudes of the AWG are offset by 2 to show them separately. Grey vertical lines are introduced to show the zero amplitude in each case. (b) Zoom in of (a) at the end of the pulses where the input amplitude was maximal. (c) Absolute values of transfer functions obtained by recording pseudo-stochastic maximum length sequences (MLS) for different AWG input amplitudes.**

the maximal output amplitudes, that it is able to achieve at lower modulations (black and red).

A more quantitative experiment to show the limitations of the AWG is shown in Fig. A2 (c), here we recorded

pseudo-stochastic maximum length sequences with the same set up and with different input amplitudes for the AWG. With

the recorded spectrometer response we determined the transfer function by the procedure described by Burkhard et al. (2023).

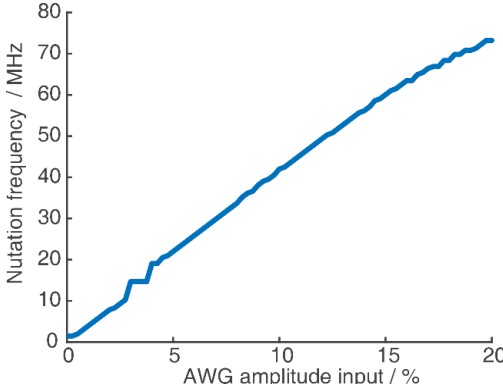

**Figure A 2: Rabi nutation amplitude profile after adjustment. A Rabi nutation experiment was recorded with different amplitudes of the nutation pulse and the nutation frequency is plotted against the AWG input amplitude. The sample here was the TEIO-N sample described in Sect. A2. The artefact at a nutation frequency of approximately 14 MHz stems from proton ESEEM contributions arising at this frequency which made it difficult to discriminate from the nutation frequency at the AWG amplitude of 4%.**





The different transfer functions were here not normalized to their maxima because we had a substantial DC leakage especially at the lower amplitudes, instead the transfer functions are normalized to their respective values at 0.075 GHz, where the DC leakage should be minimal. Ideally the transfer function of the AWG should not be amplitude dependent otherwise the AWG will not produce the same pulse shapes at different amplitudes and pulse correction with the transfer function cannot function

properly. In our set-up the shape of the transfer function was fairly constant up to an AWG input amplitude of approximately 20 % (compare Fig. A2 (c)) that is why we used the AWG only with this range in all Experiments presented in the main text. We than had to adjust the pre amplifiers in such a way that with a lower AWG input amplitude we were still able to achieve a sufficient $B_1$-field strength. The rabi nutation amplitude profile after this adjustment is shown in Fig. A3.

**A5 Pre and Post correction**

All shaped pulses used in this study were corrected with the transfer function obtained from the resonator profile by the

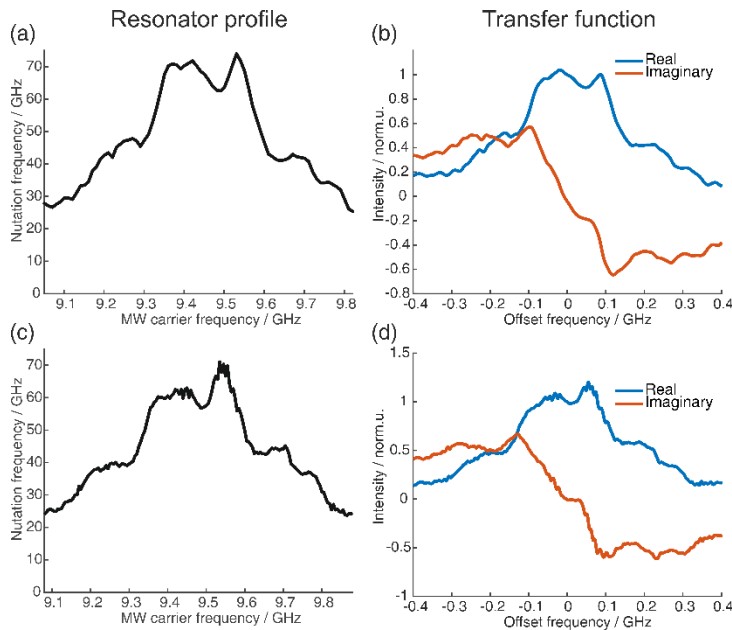

**Figure A 4: Resonator profiles and corresponding transfer functions. The resonator profile in (a) was used for the SIFTER experiments shown in Fig. 8, 9 and 10 shown in the main text. All other SIFTER experiments were recorded with the set up corresponding to (c).**

procedure described by Doll et al. (2013). The data presented in Fig. 9 and 10 were recorded in the spectrometer configuration (i.e. coupling of resonator and position of sample in resonator) whose resonator profile is shown in Fig. A4 (a). All the other SIFTER experiments were recorded with the spectrometer configuration whose resonator profile is shown in Fig. A4 (c).

In Figure A5 we show how we obtained the post-correction profile which was shown in Fig. 8 of the main text. For this we

recorded multiple SIFTER echoes at different field positions. This moves the spectrum in to different domains of the excitation bandwidth of the pulses. If both the excitation and the detection of the nitroxide would be uniform over the excitation





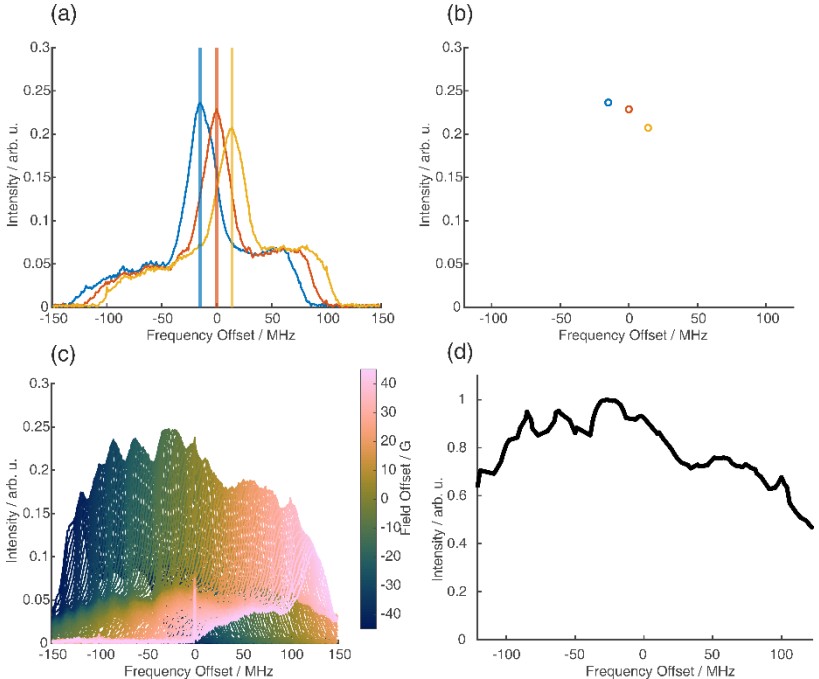

**Figure A 5: Post correction of SIFTER echo FT. (a) FT of 4 pulse-SIFTER echo with the pulse length ratio of 2-2-2-1 recorded with three different magnetic field positions (corresponding to different frequency offsets). (b) Three post-correction profile points extracted from (a). (c) FT of 4 pulse-SIFTER echo at 91 field positions from -45 to +45 G from the center field position of 3364 G. Post-correction profile obtained from following one position in the spectrum and observing its intensity as the spectrum is moved inside the excitation bandwidth of the pulses.**

bandwidth, we would expect the spectrum not to change in amplitude (as long as the spectrum stays inside the excitation bandwidth). By choosing and following any point in the spectrum and observing how its intensity is changed when the magnetic field is varied one gets a profile that includes both imperfections in excitation and refocusing of spins and of

imperfections in the detection pathway. This profile was in most cases not at all flat and there still seem to be considerable imperfections in either the excitation of the nitroxide or in the broadband detection.

**Appendix B: Phase shifts**

**B1 Parabolic phase shift**

The parabolic phase shift can be refocused by finding a combination of relative pulse lengths and sweep direction which fulfil

Eq. (8) in the main text. In general, if the pulse sequence has more pulses than different coherence pathways for which the parabolic phase shift should be refocused, Eq. (8) will result in a system of linear equations which is underdefined and has in principle infinite solutions. To find solutions for the case of Eq. (8) where there are infinite solutions one can write the left-hand side of the linear equation system as a matrix and determine the null space of this matrix. The null space spans all vectors





that satisfy the linear equation system and by expanding the basis vectors of the null space one can determine any specific

solution.

We have used this approach to write a MATLAB function which generalizes Eq. (8) and takes the coherence pathways, the position of $\pi$-pulses in the sequence, a desired maximum $Q_{crit}$ and a desired maximum integer value to which the basis vectors should be expanded and gives out a list of solutions which is sorted by how short the sequence can be made. The list also includes relative amplitudes of all the pulses and considers all possible sweep directions (for symmetric sweep directions like

upward-upward and downward-downward only one of the solutions is listed). This approach automatically determines already known solutions like the Böhlen-Bodenhausen sequence for the Hahn echo (Bohlen et al., 1989), the compressed CHORUS sequence for the refocused echo (Foroozandeh et al., 2019) and the ABSTRUSE equivalent of the stimulated echo (Jeschke et al., 2015) and can be used for any other pulse sequence and coherence pathway.

The code is made publicly available at: https://doi.org/10.25716/gude.1x8d-dh8~

**B2 Dynamic phase shifts**

In Fig. B1 we show the extend of refocusing of the dynamic phase shifts $\Delta\phi$ and $\phi_0$ in refocused chirp echo experiments (also called ABSTRUSE echo (Cano et al., 2002)). As Cano et al. discussed the two $\pi$-pulses can refocus the phase shifts that each of them introduces given that they introduce the same phase shift. Usually in multi-pulse chirp echo sequences the $SW$ between the pulses is kept constant to be able to refocus the parabolic phase shift $\phi_p$ as was discussed in the main text and in the

previous section. This was also done in the simulation shown in Fig. B1, also the value of $Q_{crit}$ between both $\pi$-pulses was kept constant. Because $\Delta\phi$ is dependent only on the $SW$ and on $Q_{crit}$ this explains why it is refocused in all cases in Fig. B1. While the $SW$ and the pulse length ratio of 2-1 between the last pulses is kept constant, this means that the two $\pi$-pulses have





different $TBP$s and therefore $\phi_0$ is not completely refocused and there is still a dependence on $Q_{crit}$ visible even at higher values of $Q_{crit}$ where phase effects because of the incomplete coherence inversion are negligible.

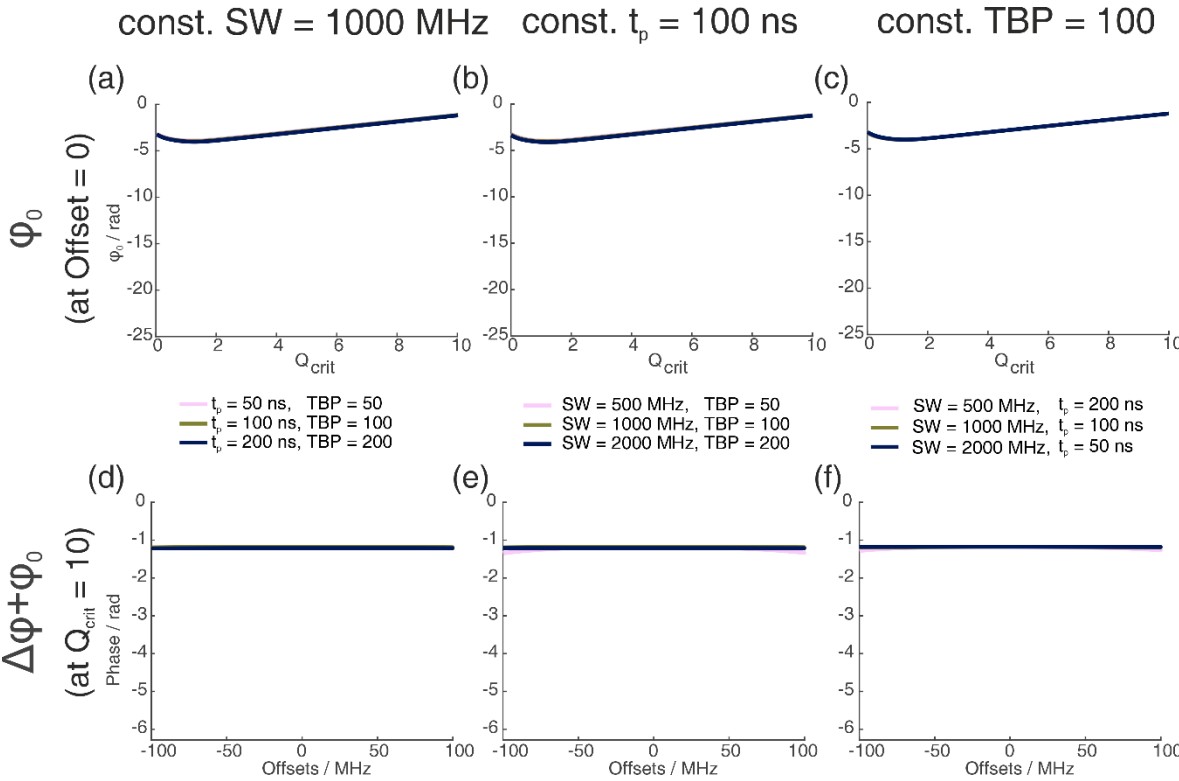

**Figure B 1: Figure 14: Numerical Bloch simulations of three pulse chirp refocused echoes with pulse length ratios of 2-2-1. The first row ((a) to (c)) show the phase $\phi_0$ at offset $\Omega = 0$ MHz as a function of $Q_{crit}$ of the second and third pulse and the second row ((d) to (f)) show the combination of $\phi_0$ and $\Delta\phi$ at $Q_{crit} = 10$ as a function of $\Omega$. For each column ((a) and (d), (b) and (e), (c) and (f)) either the sweep width $SW$, the pulse length $t_p$(shown is the length of the last pulse) or the time bandwidth product $TBP$ are held constant while the other two parameters are varied. The y-axes are for comparison scaled to the same height as the corresponding y-axes of Fig. 1 in the main text.**

## Appendix C: Additional data and analysis of SIFTER experiments

In this section we show for every SIFTER sequence a phase diagram of the refocusing of the parabolic phase shift $\phi_p$. We also show the SIFTER echo amplitude and phase when sweeping the input amplitude at the AWG of all $\pi$-pulses simultaneously. Plotted is the input amplitude of the shortest $\pi$-pulse, the amplitudes of the other $\pi$-pulses were increased so that all $\pi$-pulses should have the same $Q_{crit}$. Together with the rabi nutation amplitude profiles (data not shown) that were discussed in Sect. A4

of the appendix we calculated the corresponding values of $Q_{crit}$ from the AWG input amplitude and plotted phase and amplitude of the echo also against $Q_{crit}$. It should be noted that at high amplitudes close to 20 % the value of a $Q_{crit}$ was most likely not uniform for all $\pi$-pulses since at those amplitudes the TWT is close to saturation and the AWGs transfer function is not constant anymore (see also Sect. A4 of the appendix).



Lastly we also show the SIFTER traces before and after division of an additional exponential and the corresponding distance
distribution obtained with Tikhonov regularization in DeerAnalysis (Jeschke et al., 2006).

## C1 SIFTER with pulse length ratio 2-2-2-1

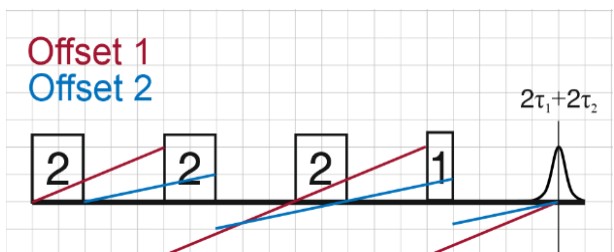

**Figure C 1: Phase diagram of a chirped 4 pulse-SIFTER sequence with a pulse length ratio 2-2-2-1. The red and blue line represent the phase gain of two spins with different Larmor frequencies that are excited at low and high frequencies respectively.**

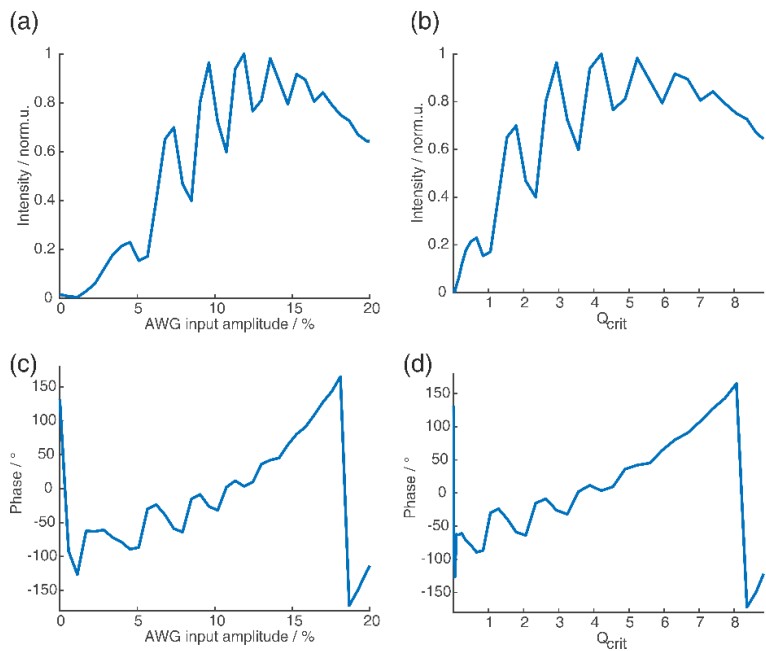

**Figure C 2: SIFTER echo vs amplitude of $\pi$-pulses. (a) Absolute intensity of the SIFTER echo as a function of AWG input amplitude of the shortest $\pi$-pulse (all other $\pi$-pulses are increased so that $Q_{crit}$ between them stays constant). (b) same as (a) but as a function of $Q_{crit}$. (c) Phase of SIFTER echo as a function of AWG input amplitude. (d) same as (c) but as a function of $Q_{crit}$.**

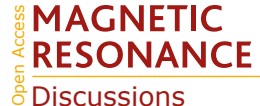

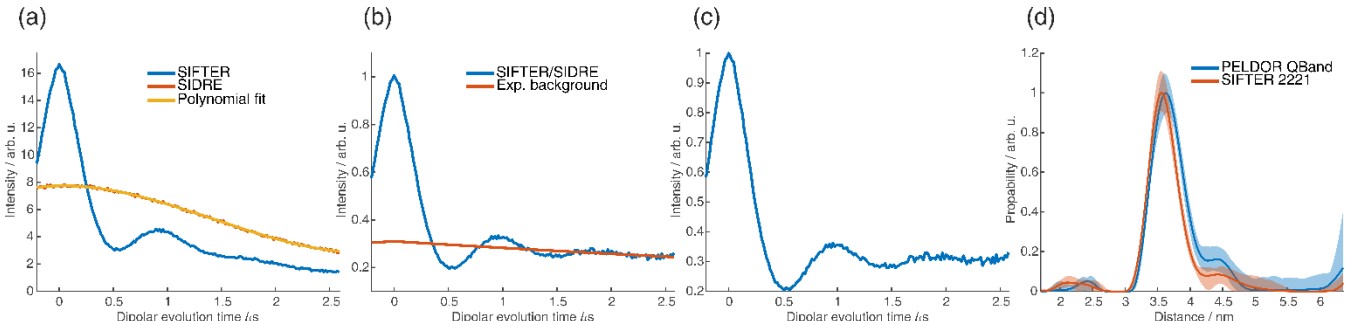

**Figure C 3: (a) 4 pulse-SIFTER (blue) and 3 pulse-SIDRE (orange) trace averaged over the whole nitroxide spectrum and a polynomial fit of the 3 pulse-SIDRE Trace. (b) 4 pulse-SIFTER data divided by polynomial fit of 3 pulse-SIDRE data (blue) and exponential background fitted by DeerAnalysis (orange). (c) Formfactor obtained by division of blue trace of (b) by orange background in (b). (d) Distance distributions obtained by Tikhonov regularisation in DeerAnalysis of PELDOR trace recorded at Q-band (blue) and of SIFTER formfactor in (c) (orange).**

**C2 SIFTER with pulse length ratio 2-2-2-2-3-2**

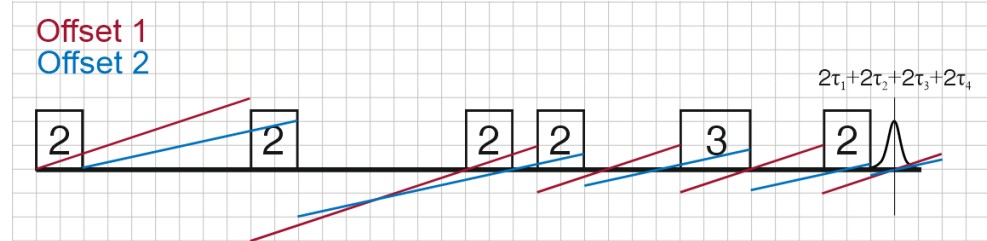

**Figure C 4: Phase diagram of a chirped 6 pulse-SIFTER sequence with a pulse length ratio of 2-2-2-2-3-2. The red and blue line represent the phase gain of two spins with different Larmor frequencies that are excited at low and high frequencies respectively.**



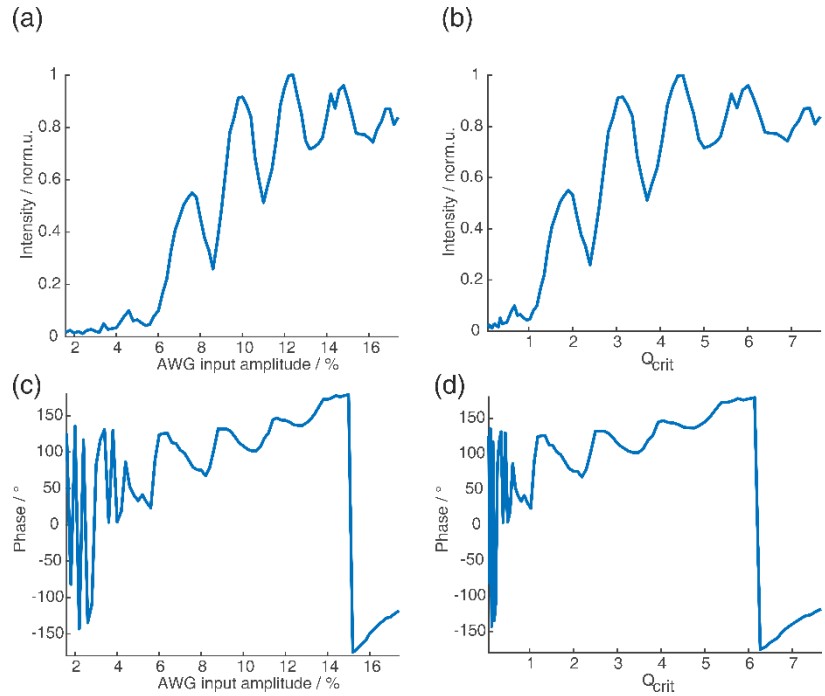

**Figure C 5: SIFTER echo vs amplitude of $\pi$-pulses for the 6 pulse-SIFTER sequence with a pulse length ratio of 2-2-2-2-3-2. (a) Absolute intensity of the SIFTER echo as a function of AWG input amplitude of the shortest $\pi$-pulse (all other $\pi$-pulses are increased so that $Q_{crit}$ between them stays constant). (b) same as (a) but as a function of $Q_{crit}$. (c) Phase of SIFTER echo as a function of AWG input amplitude. (d) same as (c) but as a function of $Q_{crit}$.**

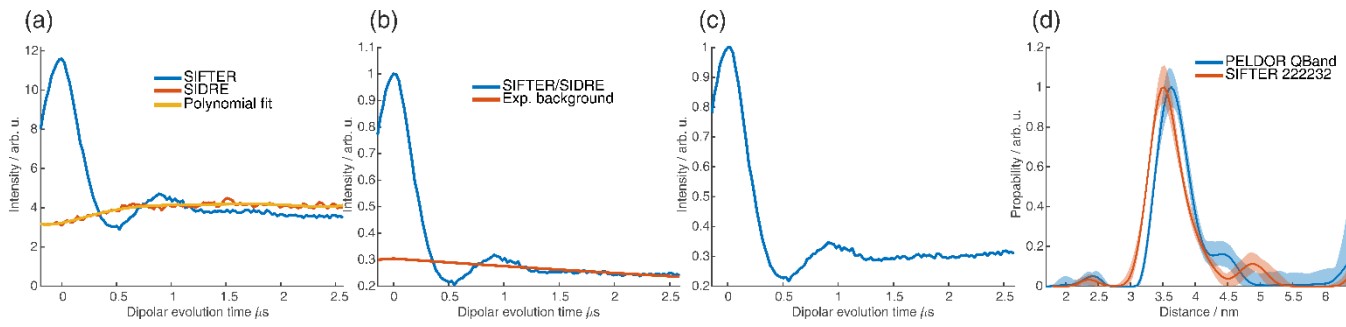

**Figure C 6: (a) 6 pulse-SIFTER (blue) and 5 pulse-SIDRE (orange) trace averaged over the whole nitroxide spectrum and a polynomial fit of the 5 pulse-SIDRE Trace. (b) 6 pulse-SIFTER data divided by polynomial fit of 5 pulse-SIDRE data (blue) and exponential background fitted by DeerAnalysis (orange). (c) Formfactor obtained by division of blue trace of (b) by orange background in (b). (d) Distance distributions obtained by Tikhonov regularisation in DeerAnalysis of PELDOR trace recorded at Q-band (blue) and of SIFTER formfactor in (c) (orange).**





## C4 SIFTER with pulse length ratio 2-3-1-4

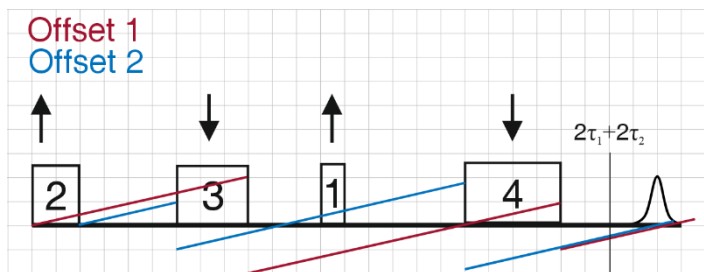

**Figure C 7: Phase diagram of a chirped 4 pulse-SIFTER sequence with a pulse length ratio of 2-3-1-4 and relative sweep directions of up-down-up-down (also indicated by arrows). The red and blue line represent the phase gain of two spins with different Larmor frequencies that are excited at low and high frequencies respectively.**

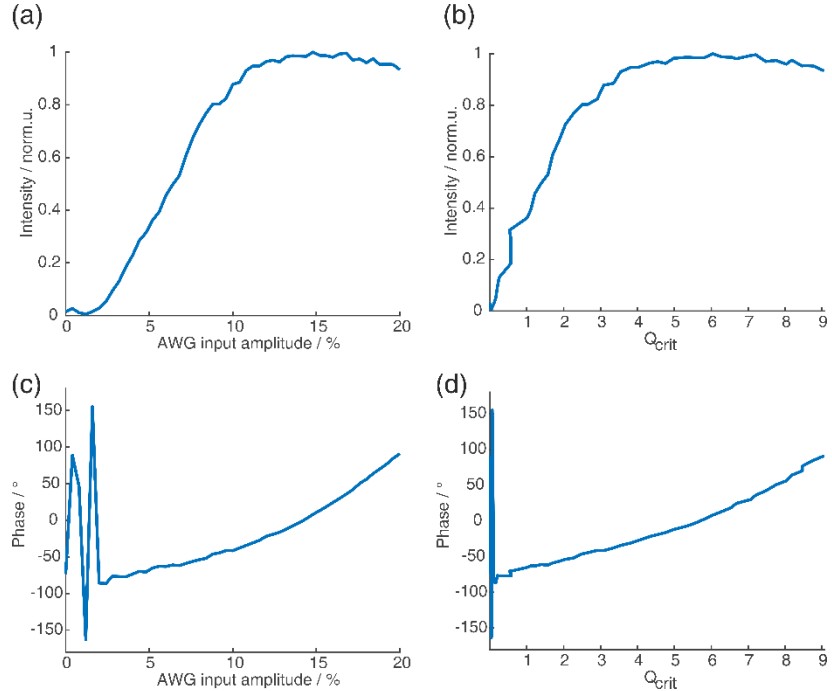

**Figure C 8: SIFTER echo vs amplitude of $\pi$-pulses for the SIFTER sequence with a pulse length ratio of 2-3-1-4. (a) Absolute intensity of the SIFTER echo as a function of AWG input amplitude of the shortest $\pi$-pulse (all other $\pi$-pulses are increased so that $Q_{crit}$ between them stays constant). (b) same as (a) but as a function of $Q_{crit}$. (c) Phase of SIFTER echo as a function of AWG input amplitude. (d) same as (c) but as a function of $Q_{crit}$.**



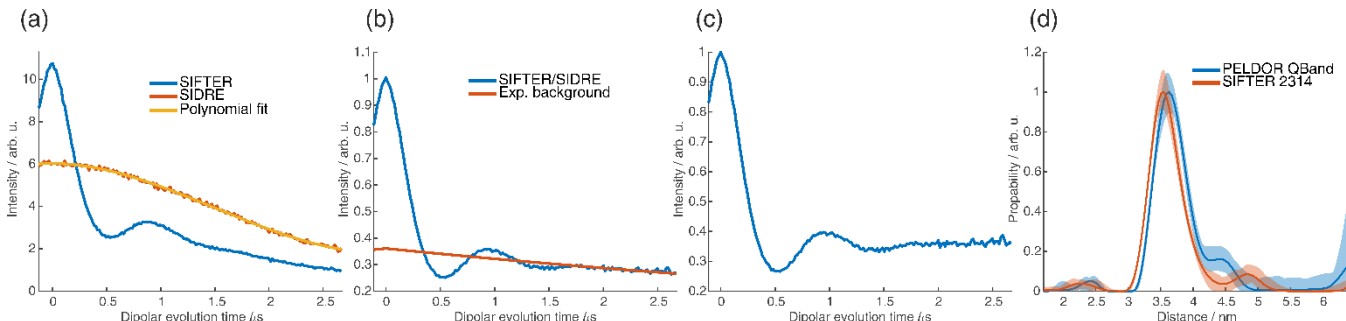

**Figure C 9: (a) 4 pulse-SIFTER (blue) and 3 pulse-SIDRE (orange) trace averaged over the whole nitroxide spectrum and a polynomial fit of the 3 pulse-SIDRE Trace. (b) 4 pulse-SIFTER data divided by polynomial fit of 3 pulse-SIDRE data (blue) and exponential background fitted by DeerAnalysis (orange). (c) Formfactor obtained by division of blue trace of (b) by orange background in (b). (d) Distance distributions obtained by Tikhonov regularisation in DeerAnalysis of PELDOR trace recorded at Q-band (blue) and of SIFTER formfactor in (c) (orange).**

**C3 SIFTER with pulse length ratio 2-2-6-3**

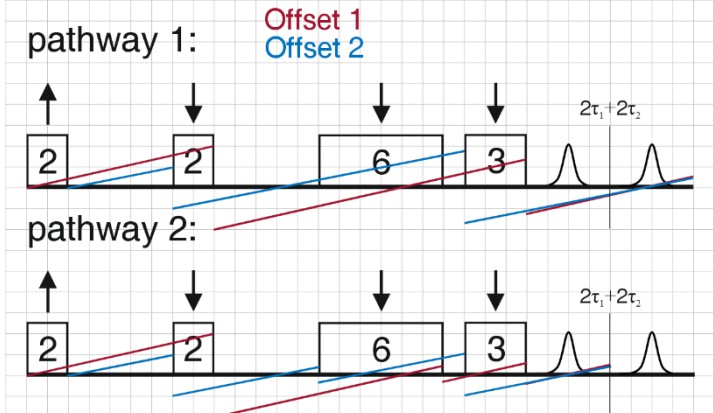

**Figure C 10: Phase diagram of a chirped 4 pulse-SIFTER sequence with a pulse length ratio of 2-2-6-3 and relative sweep directions of up-down-down-down (also indicated by arrows). The red and blue line represent the phase gain of two spins with different Larmor frequencies that are excited at low and high frequencies respectively. The sequence refocuses two coherence pathways. The typical refocused echo coherence pathway where the third pulse does not influence the evolution of the frequency offset and a second pathway the third pulse leads to coherence inversion.**





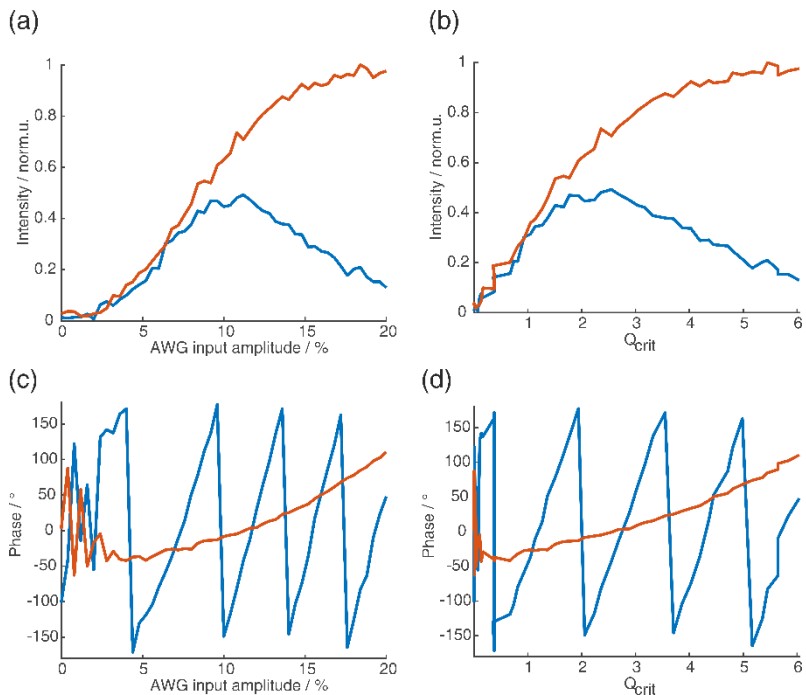

**Figure C 11: SIFTER echo vs amplitude of $\pi$-pulses for the SIFTER sequence with a pulse length ratio of 2-2-6-3. (a) Absolute intensity of the SIFTER first (blue) and second (orange) echo as a function of AWG input amplitude of the shortest $\pi$-pulse (all other $\pi$-pulses are increased so that $Q_{crit}$ between them stays constant). (b) same as (a) but as a function of $Q_{crit}$. (c) Phase of first (blue) and (second) SIFTER echo as a function of AWG input amplitude. (d) same as (c) but as a function of $Q_{crit}$.**


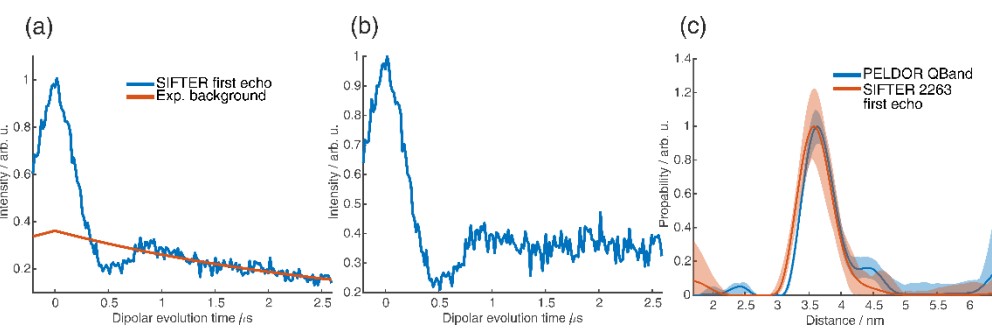

**Figure C 12: (a) 4 pulse-SIFTER of first echo in 4 pulse-SIFTER experiment with a pulse length ratio of 2-2-6-3 and sweep directions of up-down-down-down averaged over the whole nitroxide spectrum (blue) and exponential background fitted by DeerAnalysis (orange). Since the first echo only appears if third pulse is included, there is no SIDRE equivalent for the first echo. (b) Formfactor obtained by division of blue trace of (a) by orange background in (a). (c) Distance distributions obtained by Tikhonov regularisation in DeerAnalysis of PELDOR trace recorded at Q-band (blue) and of SIFTER formfactor in (b) (orange).**





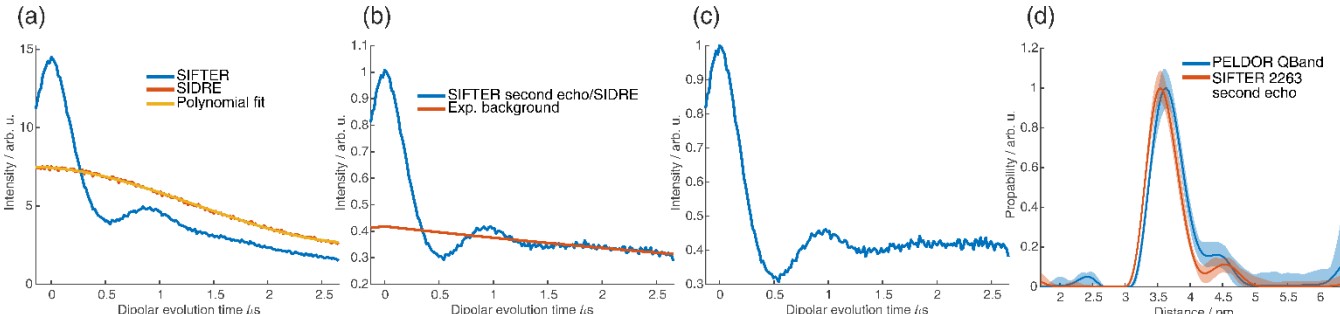

**Figure C 13: (a) 4 pulse-SIFTER (blue) and 3 pulse-SIDRE (orange) trace of second echo in SIFTER with pulse length ratios 2-2-6-3 and sweep directions of up-down-down-down averaged over the whole nitroxide spectrum and a polynomial fit of the 3 pulse-SIDRE Trace. (b) 4 pulse-SIFTER data divided by polynomial fit of 3 pulse-SIDRE data (blue) and exponential background fitted by DeerAnalysis (orange). (c) Formfactor obtained by division of blue trace of (b) by orange background in (b). (d) Distance distributions obtained by Tikhonov regularisation in DeerAnalysis of PELDOR trace recorded at Q-band (blue) and of SIFTER formfactor in (c) (orange).**

## Appendix D: EPR-correlated SIFTER data

Here we show the EPR-correlated SIFTER data sets of the SIFTER experiments not shown in the main text. The data sets

were obtained by performing, additionally to the complex direct FT of the SIFTER echo, a magnitude FT along the dipolar evolution time axis. Before the magnitude FT along the dipolar time axis was performed, we carried out a background correction procedure. For all 4-pulse experiments we first divided each dipolar time trace slice by a polynomial fit of the corresponding 3-pulse SIDRE trace and then divided the resulting trace by an additional exponential that we fit to the end of that trace. For the 6-pulse SIFTER experiment we omitted the division of the 5-pulse trace. We then normalized each slice and

multiplied it by the original intensity before the background procedure. Finally, we performed a magnitude FT of the dipolar time axis without any zero filling or apodization. Our background procedure did not completely remove the dipolar zero frequency artefact especially at the edges of the EPR spectrum were noise made fitting of the exponential more difficult. To visualize the data, we show for each dataset a contour plot where the contour lines are logarithmically spaced and a heatmap where we normalized each slice along the dipolar frequency axis and squared the values, to facilitate visually following the

main peaks in the dipolar spectrum. In the case of the 4 pulse-SIFTER sequence with pulse length ratio 2-2-6-3 only the second echo was analysed, as the first echo exhibited an insufficient SNR. To prevent overlap between the two echoes during the analysis of the second echo, we truncated the transient, which led to a less well-resolved offset frequency axis compared to the other experiments.




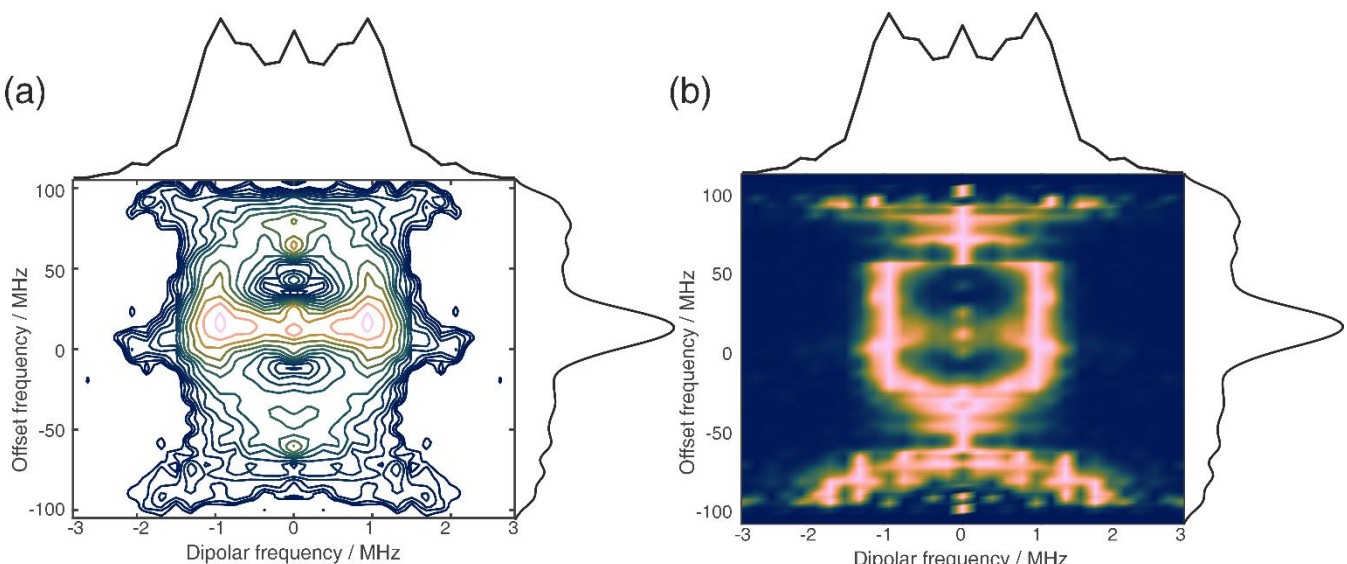

**Figure D 1: 2D EPR-correlated SIFTER data of second echo in the 4 pulse-SIFTER experiment with a pulse length ratio 2-2-6-3 and sweep directions up-down-down-down. The background correction procedure is described in the text in Sect. D. The dipolar spectrum shown above the correlation plots is the sum of all dipolar spectra and the EPR spectrum shown on the side is the real part of the direct FT at the maximum of the time trace. (a) contour plot of 2D FT of SIFTER echoes with logarithmically spaced contour lines. (b) heatmap of 2D FT of SIFTER echoes where each dipolar spectrum was normalized and squared to facilitate visually following the main peaks in the dipolar spectrum.**

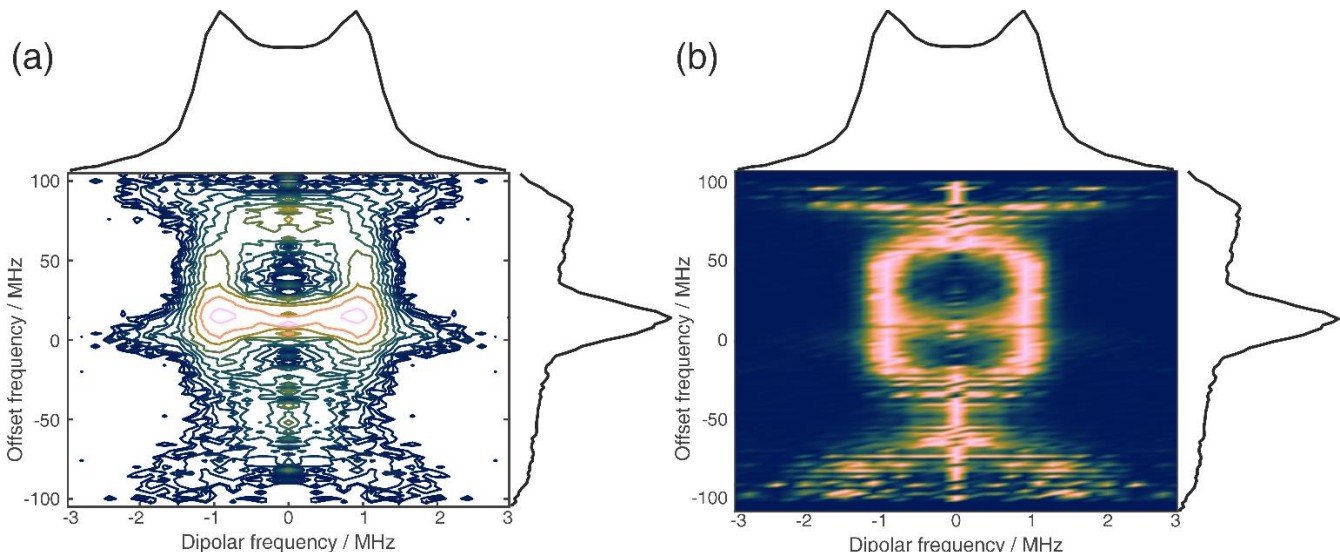

**Figure D 2: 2D EPR-correlated SIFTER data of the 4 pulse-SIFTER experiment with a pulse length ratio of 2-3-1-4 and sweep directions up-down-up-down. The background correction procedure is described in the text in Sect. D. The dipolar spectrum shown above the correlation plots is the sum of all dipolar spectra and the EPR spectrum shown on the side is the real part of the direct FT at the maximum of the time trace. (a) contour plot of 2D FT of SIFTER echoes with logarithmically spaced contour lines. (b) heatmap of 2D FT of SIFTER echoes where each dipolar spectrum was normalized and squared to facilitate visually following the main peaks in the dipolar spectrum.**


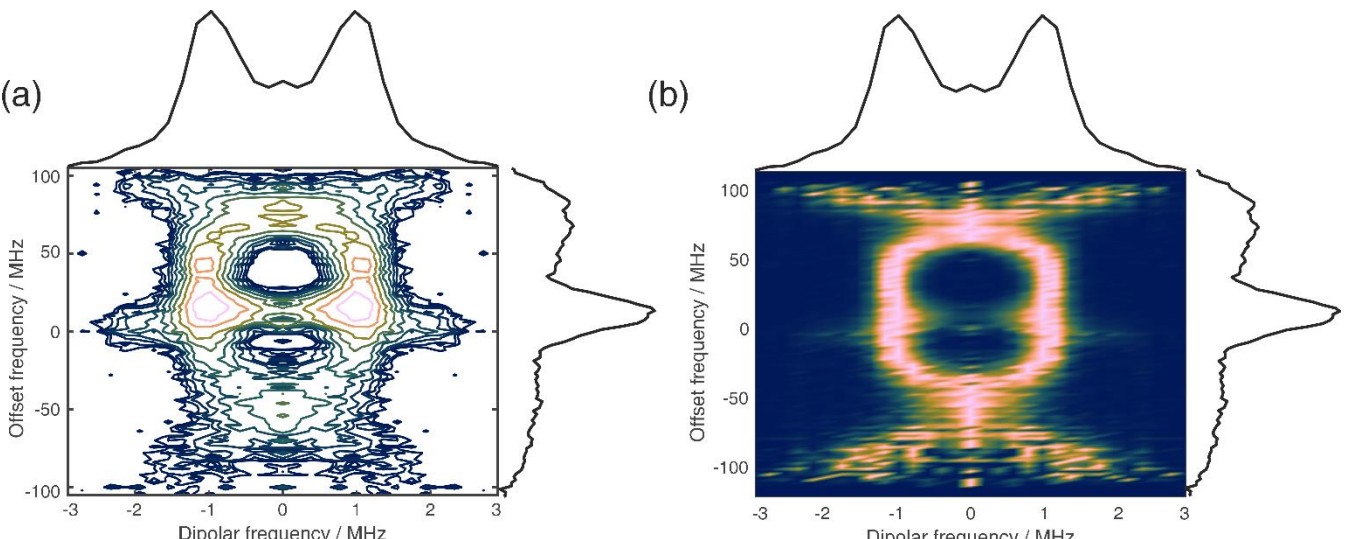

**Figure D 3: 2D EPR-correlated SIFTER data of the 6 pulse-SIFTER experiment with a pulse length ratio 2-2-2-2-3-2 and sweep directions of only upsweeps. The background correction procedure is described in the text in Sect. D. The dipolar spectrum shown above the correlation plots is the sum of all dipolar spectra and the EPR spectrum shown on the side is the real part of the direct FT at the maximum of the time trace. (a) contour plot of 2D FT of SIFTER echoes with logarithmically spaced contour lines. (b) heatmap of 2D FT of SIFTER echoes where each dipolar spectrum was normalized and squared to facilitate visually following the main peaks in the dipolar spectrum.**

## Code and data availability

The experimental data and the scripts for finding different pulse ratios to refocus the parabolic phase shift are available at the Goethe University Data Repository under https://doi.org/10.25716/gude.1x8d-dh8~

## Author contribution

PT, BE and TP designed the research. The sample was prepared in the lab from SS. Experiments were carried out by PT with help from BE. PT carried out the data analysis and simulations. The article was written by PT and edited by BE, SS and TP.

## Competing interests

TP is a member of the editorial board of Magnetic Resonance.



**Acknowledgements**

Dr. Maximilian Gauger prepared the EPR sample of the RNA duplex and performed the PELDOR measurements with gaussian pump and probe pulses on this sample which are shown here for comparison. Dr. Anna-Lena J. Halbritter from the Snorri Sigurdsson group performed the synthesis of the RNA duplex with rigid spin labels attached. We used the colormaps provided from (Crameri et al., 2020).

**Financial support**

We thank the BMRZ for financial support of the EPR infrastructure and the DFG for funding of this project.

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
