# Peer review of "Optimized shaped pulses for 2D-SIFTER"

_Magnetic Resonance, 2025_

## Referee Comment (RC1)

In this study, Trenkler et al. discuss and show the use of shaped pulses (chirps in this case) for SIFTER. The broadband excitation by these pulses allows the "direct" detection of the whole EPR spectrum for each time point in the SIFTER trace, via a Fourier transform of the echo. This correlates the dipolar coupling in the indirect dimension with the EPR spectrum in the direct dimension. Since the EPR spectrum of nitroxides at X-band essentially encodes orientational information, the 2D correlation in these experiments allows to determine not only the distance between two labels, but also their orientation.

The use of shaped pulses in SIFTER and the dipolar/EPR correlation is not new. Chirped SIFTER was shown in (Schöps et al., 2015), albeit without a direct FT dimension. The dipolar/EPR correlation was demonstrated by (Doll and Jeschke, 2016) in Q-band, and by (Bowen et al., 2018).

Nevertheless, the present study discusses many theoretical and practical intricacies in a clear and nice way, and the application to rigid labels in RNA plus the comparison with orientation-selective DEER/PELDOR is certainly interesting to many researchers in the field. I also believe that the influence of  $Q_{crit}$  on  $\phi_0$  was not discussed so clearly before (at least I never thought about it in detail). This doesn't matter in many sequences, but in SIFTER it does. This should also have implications for other sequences, such as 5p-ESEEM.

Given these points, I think the paper is a valuable contribution, and certainly well suited for *Magnetic Resonance*.

I have a few points which should be considered for a revised version, and then I have some questions about the hardware. The latter is not the focus of this paper, but I ask anyway because I am interested.

- 1) The chirp-SIFTER papers above are already cited in the manuscript, but given the title, I think it would be appropriate to mention them in the introduction already.
- 2) Line 16: I think "orientation selective SIFTER" is a bit unfortunate in this context. I know where the author's come from, but the particular experiment is as orientation NON-selective as it gets. I would use "EPR-correlated", or "Orientation-correlated", although the latter is a bit less accurate.
- 3) Line 23: I think the nuclear spin with the largest magnetic moment is (3H);)
- 4) Line 46: "However, for shaped pulses it is essential to keep all amplifiers in their linear regime, to preserve the designed pulse shape at all power levels."
  - I disagree. At X- and Q-band where there are true amplifiers and not amplifiermultiplier chains, the power amplifier can be driven into saturation. Especially for chirps, the amplitude non-linearity is not very detrimental, unless the input

- signal is not clean. If LO leakage is significant, the non-linearity will introduce harmonics at  $\omega_{LO}+n\omega_{awg}$ , which can be very bad for low AWG frequencies. I do agree however, that the non-linearity must be taken into account, especially for amplitude modulation. In commercial NMR spectrometers, this is routinely done, in a process called "linearization".
- 5) In the definition of the WURST pulse as given in the paper, the phase at t=0 (middle of the pulse) is not 0, but  $\phi(t) = \int_{t_p/2}^t \Delta\omega(t') \mathrm{d}t' = \pi SW \left(\frac{t^2}{tp} \frac{tp}{4}\right)$ , so  $\phi(0) = -\frac{1}{4}\pi \, SW \, t_p$ . This is the point where it crosses the spins at 0 offset and contains the TBP. I would just like the authors to confirm that the particular definition has no influence on their discussion regarding  $Q_{crit}$ ,  $\phi_0$  and TBP.
- 6) Line 267: "It also needs to be verified that the output of the AWG (arbitrary waveform generator) is linear with respect to the input amplitudes for fast amplitude changes."
  - I would just like to highlight that the Nyquist criterion alone is not sufficient to talk about the bandwidth of an AWG or a digitizer. While the authors use an AWG with 0.625 ns sampling step, the info I found online also mentioned an analog bandwidth of 400 MHz, which is exactly what the authors see in Figure A 3. In fact, the analog bandwidth can be smaller (oversampling) or higher (undersampling) than the Nyquist frequency.
- 7) Line 685: "All shaped pulses used in this study were corrected with the transfer function obtained from the resonator profile by the procedure described by Doll et al. (2013)." Is unclear to me. Did you get the transfer function from the resonator profile as A. Doll did, or did you correct the pulses like Doll, or both? The "correction" could mean a) deconvolution with the transfer function, or b) adjusting the sweep rate of the chirp according to the resonator profile.
- 8) Line 695: "This profile was in most cases not at all flat and there still seem to be considerable imperfections in either the excitation of the nitroxide or in the broadband detection". It is expected that the detection is influenced by the resonator profile. It has to be, and this cannot be compensated in the excitation.
- 9) Line 277: "In both spectrometers, there was no clock synchronization of the spectrometer pulses and transient recorder to the AWG. In the case of our homebuilt spectrometer, since the oscilloscope has a much higher time resolution (0.1 ns compared to 0.625 ns of AWG), this was not a concern."
  - I find this puzzling. I don't see how the higher time resolution of the oscilloscope solves the syncing problem, unless you digitally phase-correct each transient. I can see that for synchronized triggers and low frequency (IF) signals, this might not matter so much. But I would expect that for the higher frequencies in the nitroxide spectrum, averaging echoes with varying phase should lead to some attenuation. Did you verify that echoes average coherently over many shots and over longer time periods?

- 10) Line 359: "Importantly, 360 the great agreement between the SIFTER echo FT and the EDFS was achieved only after identifying, through initial testing, a spectrometer carrier frequency at which standing waves did not significantly distort the microwave pulse shape or the detection."
  - I suspect not only standing waves, but also mixing artifacts/spurious frequencies, due to the IQ mixing with the LO in the middle of the resonator and spectrum.
- 11) Line 414: The echoes in orientation-selective DEER/PELDOR could also be Fourier-transformed.

---

## Referee Comment (RC2)

Review "Optimized shaped pulses for 2D-SIFTER"

The paper by Paul Trenkler et al presents a great and comprehensive discussion of a number of different effects with respect to chirp SIFTER, in particular a comprehensive discussion of the different phase effects known to occur in chirp echo sequences. Further the authors provide an impressive, careful comparison of SIFTER and PELDOR data on the same RNA duplex, showing virtually quantitative agreement in a fraction of the data acquisition time by virtue of the broadband excitation, which is a great result. While many aspects, including chirp SIFTER, are known already as cited by the authors, the paper is definitely worth publishing in Magnetic Resonance as it goes beyond the current literature and enhances the applicability of chirp SIFTER, not least with the additional SIFTER pulse sequences. In particular, the different phase effects are comprehensively summarized, supported by simulations, and the importance for sequences like SIFTER is clearly explained, that this aspect could even be mentioned in abstract & conclusions.

To further improve clarity, I have the following comments prior to publication:

To show the complete SIFTER data, also the SIFTER raw data for the background-corrected traces in Fig. 10 a) need to be included somewhere as individual traces (ideally stacked like in Fig. 10a), particularly in the light of the uncertainties associated with the background correction (see below).

Discussion and simulations of optimal chirp pi-pulse amplitudes in Fig. 3 are great in their illustrative detail and show clearly why it was important to optimize chirp pulse amplitudes carefully as described in previous studies. Here, in the ideal simulations in the absence of B1-inhomogeneity, for a clear picture it would be important to state why the "Fidelity" already declines without the B1-inhomogeneity. Further, what is the direct consequence of the increasing B1-inhomogeneity, which causes the observed echo decline at high Qcrit? Is it a more severe phase roll delta\_phi? This is assuming that the fixed echo phases phi\_0 have been corrected for each Qcrit. Is that the case? In the text (p. 10) it should be "small sample", rather than "large sample" (the stated color is correct).

In relation to the nice simulations in Fig. 4, p. 10, the term "destructive interference pattern" is unclear. Also, please confirm/state in the text clearly that all sequences in Fig. 4 are set up in a way that complies with Eq. (8), as briefly indicated in the figure caption.

In line with the comment of Nino Wili, it should be correctly explained that working in the linear regime of the microwave amplifiers is not a requirement, contrarily to the authors' statement. It should be discussed that amplifier non-linearity or compression should be quantified and accounted for when calculating the microwave pulse shapes, or otherwise (when on purpose avoided for model studies) significant microwave power needs to be sacrificed to remain in the linear regime of a TWT amplifier. To avoid the caveat of sacrificing microwave power, software compensation with experimental amplitude compression appears preferable.

For the instrumentation, in the absence of high-speed clocks, it would be better to report a jitter time for single echoes (1 shot) for the standard and the high-speed detection variants because without high-speed clock synchronization there is also no fixed phase relation. I strongly suspect, that for down-conversion to 0 GHz before the ADC a small remaining jitter should have indeed only marginal influence (if any) — however, chirp echoes will include higher frequency components, hence jitter should be known before averaging shots to quantify echo amplitudes.

In agreement with the public comment of Maxim Yulikov, I share his opinion that the discussion of the SIFTER background and particularly of its correction here falls short of the potential offered by the presented data. It is a great first step that experimental SIDRE traces have been obtained, and surely

also a 2-pulse Hahn echo decay should be known for these samples. Now that a strategy for background correction is published, as cited by Vanas et al 2023, it appears adequate to at least discuss why the presented approximation was chosen and explain how this could be justified over obtaining the SIFTER form factor F(t) with  $F(t) = [S\_SIFTER(t) - S\_unmodulated(t)] / [S\_SIDRE(t) + S\_2Hahn(t)]$ , where S are the different time domain signals and  $S\_2Hahn(t)$  is the decay of 2 Hahn echoes multiplied back-to-back as described by Vanas et al 2023. The approach taken here and the resulting quantitative agreement with the PELDOR data, suggests that the  $S\_2Hahn(t)$  signals here may have a form very similar to the  $S\_SIDRE$  signal for the length of the SIFTER traces shown here. These points appear important to test and clarify in order to provide solid grounds for the background correction employed here.

While indeed Chirp echoes are not the current experimental standard, the literature is somewhat less sparse than it sounds in the introduction. In addition to the works already cited, there are a number of notable applications optimizing and using chirp echoes, such as ESEEM (DOI 10.1063/1.4927088), chirp RIDME (doi:10.2533/chimia.2019.268), and CHEESY-detected EDNMR (DOI 10.1016/j.jmr.2018.02.001 & DOI: 10.1039/DOSC04436A). Also, experimental characterization of pulse excitation profiles has been shown and used as calibration (e.g. Fig. 6 b/e in DOI: 10.1039/c7cp01488k).

In the discussion of Fig. 2 it could be made clear that the sequence with pulse length ratios of 2-2-2 does not satisfy Eq.(8), while the ratio of 2-2-1 does. The expected result is the pronounced phase roll observed in panel (b) (for which the fig. captions is incomplete). The expectation based on Eq. (8) together with this result could be discussed more clearly.

p2 When discussing instrumental non-linearities, Ilya Kuprov's GRAPE paper (doi: 10.1063/5.0264092) could be cited as solution here to be aware of and discuss the approach to compensate for hardware imperfections in software.

p2 When other signals (here called unwanted) of SIFTER or DQC are discussed, it should be noted that most other coherence transfer pathways are not deleterious anymore, but instead can be explicitly taken into account during data analysis in DeerLab using multi-pathway fitting.

p3 As mentioned (and known, see e.g. Verstraete et al 2021) there is a difference between the nominal setting and the actual effective excitation bandwidth of chirp pulses. How much is that difference under the conditions used here? This is e.g. experimentally quantified in pulse excitation profiles.

p4 The finding that smaller Qcrit values (with large TBP values) for pi-pulses can lead to larger echo amplitudes is also discussed in the Verstraete 2021 paper. With the minimum value of TBP = 30 for pi-pulses, it would be helpful to also mention suitable values for the pi/2-pulses used here.

p6 While defining the different phases, please also define phi\_p the parabolic phase shift, and for clarity please confirm that within your definitions that the Bloch Siegert shift and the term "dynamic phase shifts" are to be considered equal.

e.g. p9: Please don't omit "chirp" in 2-pulse chirp Hahn echo for clarity, as many might understand the conventional Hahn echo to have rectangular pulses.

p12 and appendix: What is an AWG input amplitude, considering that the AWG is a microwave source?

p16, Fig. 8a it would be nice to mark the 0 MHz offset point in the resonator profile for the other panels in Fig 8.

p17 In the brief introduction of orientation-selective PELDOR, particularly here with respect to X-band, also DOI: 10.1016/j.jmr.2011.12.024 would be adequate to cite.

p19 "SIFTER does not have these limitations and the frequency resolution is only limited by the homogeneous linewidth and the SNR..." Why homogeneous rather than inhomogeneous linewidth here? The EPR spectrum consists of a large number of inhomogeneously broadened EPR lines, some of which are excited at each frequency step during a chirp pulse.

p19 ".. dipolar frequencies from omega\_dd to 2 omega\_dd at the edges of the spectrum." Fig. 10 a looks like omega\_dd is observed at the two edges, whereas 2 omega\_dd is observed in the center (around 0 MHz offset in Fig. 10).

p26, Fig. 15 If a SIDRE trace is available as reference to the 6-pulse SIFTER, it would be nice to also show this in Fig. 15.

---

## Author Comment (AC1)

**Response to referee-1 Nino Willi**

In this study, Trenkler et al. discuss and show the use of shaped pulses (chirps in this case) for SIFTER. The broadband excitation by these pulses allows the "direct" detection of the whole EPR spectrum for each time point in the SIFTER trace, via a Fourier transform of the echo. This correlates the dipolar coupling in the indirect dimension with the EPR spectrum in the direct dimension. Since the EPR spectrum of nitroxides at X-band essentially encodes orientational information, the 2D correlation in these experiments allows to determine not only the distance between two labels, but also their orientation.

The use of shaped pulses in SIFTER and the dipolar/EPR correlation is not new. Chirped SIFTER was shown in (Schöps et al., 2015), albeit without a direct FT dimension. The dipolar/EPR correlation was demonstrated by (Doll and Jeschke, 2016) in Q-band, and by (Bowen et al., 2018).

Nevertheless, the present study discusses many theoretical and practical intricacies in a clear and nice way, and the application to rigid labels in RNA plus the comparison with orientation-selective DEER/PELDOR is certainly interesting to many researchers in the field. I also believe that the influence of Qcrit on  $\phi0$  was not discussed so clearly before (at least I never thought about it in detail). This doesn't matter in many sequences, but in SIFTER it does. This should also have implications for other sequences, such as 5p-ESEEM.

Thank you for the positive comments! Indeed the  $\phi_0$  phase shift also plays a dominant role for the decay of the chirp-Hahn echo signal with increasing pulse amplitude under B1-inhomogeneity. Additionally, this effect will also be important for example for performing MQ-filtered EPR.

Given these points, I think the paper is a valuable contribution, and certainly well suited for *Magnetic Resonance*.

I have a few points which should be considered for a revised version, and then I have some questions about the hardware. The latter is not the focus of this paper, but I ask anyway because I am interested.

1) The chirp-SIFTER papers above are already cited in the manuscript, but given the title, I think it would be appropriate to mention them in the introduction already.

**Response:**

Thank you for this comment we have done this.

2) Line 16: I think "orientation selective SIFTER" is a bit unfortunate in this context. I know where the author's come from, but the particular experiment is as orientation NON-selective as it gets. I would use "EPR-correlated", or "Orientation-correlated", although the latter is a bit less accurate.

**Response:**

Thank you. In a revised version we will consistently use the term "EPR-correlated SIFTER".

3) Line 23: I think the nuclear spin with the largest magnetic moment is (3H);)

**Response:**

You are correct, Tritium indeed has a slightly higher magnetic moment. However, since 3H is an unstable isotope and rarely used in NMR we will keep referencing the proton for simplicity.

"Since the electron spin's magnetic moment is almost three orders of magnitude larger than that of the proton (1H), EPR offers intrinsically much higher sensitivity."

4) Line 46: "However, for shaped pulses it is essential to keep all amplifiers in their linear regime, to preserve the designed pulse shape at all power levels." I disagree. At X- and Q-band where there are true amplifiers and not amplifier-multiplier chains, the power amplifier can be driven into saturation. Especially for chirps, the amplitude non-linearity is not very detrimental, unless the input

signal is not clean. If LO leakage is significant, the non-linearity will introduce harmonics at  $\omega_{LO}+n\omega_{awg}$ , which can be very bad for low AWG frequencies. I do agree however, that the non-linearity must be taken into account, especially for amplitude modulation. In commercial NMR spectrometers, this is routinely done, in a process called "linearization".

Thank you for your comment You are correct, driving the amplifier almost to saturation and applying linearization afterwards is also a valid mode of operation that will give higher peak  $B_1$ -strengths, as we have also shown in Endeward et al. (2023). We were worried about potential phase drifts of the TWT close to saturation and opted to work in the linear regime. But we will mention the other approach in the revised version.

5) In the definition of the WURST pulse as given in the paper, the phase at t=0 (middle of the pulse) is not 0, but  $\phi(t) = \int_{t_p/2}^t \Delta\omega(t') dt' = \pi SW\left(\frac{t^2}{t_p} - \frac{t_p}{4}\right)$ , so  $\phi(0) = -\frac{1}{4}\pi SW t_p$ . This is the point where it crosses the spins at 0 offset and contains the TBP. I would just like the authors to confirm that the particular definition has no influence on their discussion regarding  $Q_{crit}$ ,  $\phi_0$  and TBP.

**Response:**

Thank you we have adjusted the equation accordingly! The definition is indeed missing a constant offset  $-\phi(0)$ , as the phase in the center of the pulse was always set to zero. The correct definition should therefore be  $\phi(t) = \int_{-t_p/2}^t \Delta\omega(t')dt' - \phi(0)$ . The definition was correctly implemented in all our simulations and experiments and was only missing in the written expression. Therefore, it has no influence on our discussion of  $Q_{crit}$ ,  $\phi_0$  and TBP.

6) Line 267: "It also needs to be verified that the output of the AWG (arbitrary waveform generator) is linear with respect to the input amplitudes for fast amplitude changes." I would just like to highlight that the Nyquist criterion alone is not sufficient to talk about the bandwidth of an AWG or a digitizer. While the authors use an AWG with 0.625 ns sampling step, the info I found online also mentioned an analog bandwidth of 400 MHz, which is exactly what the authors see in Figure A 3. In fact, the analog bandwidth can be smaller (oversampling) or higher (undersampling) than the Nyquist frequency.

**Response:**

Thank you for the comment. We believe that it might be the maximum current change limiting these devices for fast and large amplitude modulations, because if it would be the bandwidth, it should also give a reduced amplitude for smaller amplitude variations with the triangle. Independent of the source of this effect it will influence the pulse shape and needs to be considered setting up shaped pulses to avoid pulse distortions. We have observed this for three different AWGs, including the AWG used in the commercial pulsed EPR setup. Therefore, we changed the wording in the manuscript.

7) Line 685: "All shaped pulses used in this study were corrected with the transfer function obtained from the resonator profile by the procedure described by Doll et al. (2013)." Is unclear to me. Did you get the transfer function from the resonator profile as A. Doll did, or did you correct the pulses like Doll, or both? The "correction" could mean a) deconvolution with the transfer function, or b) adjusting the sweep rate of the chirp according to the resonator profile.

**Response:**

We did (a), a deconvolution with the transfer function that we obtained from the resonator profile.

8) Line 695: "This profile was in most cases not at all flat and there still seem to be considerable imperfections in either the excitation of the nitroxide or in the broadband detection". It is expected that the detection is influenced by the resonator profile. It has to be, and this cannot be compensated in the excitation.

**Response:**

That is correct however, we cannot distinguish whether the distortion arises from imperfections in the transfer function or from distortion in the detection or both. For this reason, we have mentioned both possibilities here.

9) Line 277: "In both spectrometers, there was no clock synchronization of the spectrometer pulses and transient recorder to the AWG. In the case of our home-built spectrometer, since the oscilloscope has a much higher time resolution (0.1 ns compared to 0.625 ns of AWG), this was not a concern." I find this puzzling. I don't see how the higher time resolution of the oscilloscope solves the syncing problem, unless you digitally phase-correct each transient. I can see that for synchronized triggers and low frequency (IF) signals, this might not matter so much. But I would expect that for the higher frequencies in the nitroxide spectrum, averaging echoes with varying phase should lead to some attenuation. Did you verify that echoes average coherently over many shots and over longer time periods?

**Response:**

With the oscilloscope we are indeed using synchronized triggers and have not observed any significant phase changes in between single shot experiments, not even at the higher relevant frequencies. We changed the sentence accordingly to avoid misunderstanding.

10) Line 359: "Importantly, the great agreement between the SIFTER echo FT and the EDFS was achieved only after identifying, through initial testing, a spectrometer carrier frequency at which standing waves did not significantly distort the microwave pulse shape or the detection." I suspect not only standing waves, but also mixing artifacts/spurious frequencies, due to the IQ mixing with the LO in the middle of the resonator and spectrum.

**Response:**

We could minimize mixing artifacts/spurious frequencies in our home build set-up by applying DC-offsets to the I and Q ports of the modulator. In our implementation these offsets are provided by the digital to analogue converters of the AWG itself. Additionally, we employ a switch which opens only during the pulses to reduce LO leakage if the AWG pulse shape also contains delays. The calibration of the DC offsets must indeed be carried out carefully and is performed each time the LO frequency is changed, as otherwise the resulting LO leakage can become very significant. We added a comment in the revised manuscript.

11) Line 414: The echoes in orientation-selective DEER/PELDOR could also be Fourier-transformed.

**Response:**

While technically one can Fourier-transform the echo of a rectangular pulse, the additional frequency resolution gained by this would be minimal. The excitation profile of a rectangular pulse gives only good SNR at the center frequency of the pulse. Fourier transforming the echoes of rectangular PELDOR/DEER has therefore not been common practice.

**Response to referee-2 Daniel Klose**

Review "Optimized shaped pulses for 2D-SIFTER"

The paper by Paul Trenkler et al presents a great and comprehensive discussion of a number of different effects with respect to chirp SIFTER, in particular a comprehensive discussion of the different phase effects known to occur in chirp echo sequences. Further the authors provide an impressive, careful comparison of SIFTER and PELDOR data on the same RNA duplex, showing virtually quantitative agreement in a fraction of the data acquisition time by virtue of the broadband excitation, which is a great result. While many aspects, including chirp SIFTER, are known already as cited by the authors, the paper is definitely worth publishing in Magnetic Resonance as it goes beyond the current literature and enhances the applicability of chirp SIFTER, not least with the additional SIFTER pulse sequences. In particular, the different phase effects are comprehensively summarized, supported by simulations, and the importance for sequences like SIFTER is clearly explained, that this aspect could even be mentioned in abstract & conclusions.

We thank you for your comments and will address the individual questions in the following separately. Thank you also for acknowledging our explanation of the different phase effects; we will also mention our discussion of the different phase effects of chirped SIFTER already in abstract and conclusion of the revised manuscript.

To further improve clarity, I have the following comments prior to publication:

To show the complete SIFTER data, also the SIFTER raw data for the background-corrected traces in Fig. 10 a) need to be included somewhere as individual traces (ideally

stacked like in Fig. 10a), particularly in the light of the uncertainties associated with the background correction (see below).

**Response:**

We will include a plot of the stacked raw traces, and the stacked traces after division of the SIDRE trace and after division of the additional exponential in the appendix in the revised version. See Fig. C4 and C5 in the revised version.

Discussion and simulations of optimal chirp pi-pulse amplitudes in Fig. 3 are great in their illustrative detail and show clearly why it was important to optimize chirp pulse amplitudes carefully as described in previous studies. Here, in the ideal simulations in the absence of B1-inhomogeneity, for a clear picture it would be important to state why the "Fidelity" already declines without the B1-inhomogeneity. Further, what is the direct consequence of the increasing B1-inhomogeneity, which causes the observed echo decline at high Qcrit? Is it a more severe phase roll delta\_phi? This is assuming that the fixed echo phases phi\_0 have been corrected for each Qcrit. Is that the case? In the text (p. 10) it should be "small sample", rather than "large sample" (the stated color is correct).

**Response:**

Thank you for the comments. As you address multiple points here, we will answer them separately:

Point 1: Here, in the ideal simulations in the absence of B1-inhomogeneity, for a clear picture it would be important to state why the "Fidelity" already declines without the B1-inhomogeneity

The decline in "Fidelity" in the absence of B1-inhomogeneity can be explained by  $\Delta\phi(\Delta\omega,Q_{crit})$ , which introduces different phase shifts for different offsets  $\Delta\omega$ , even without any B1-inhomogeneity. As the variance of  $\Delta\phi$  across offsets increases with  $Q_{crit}$ , destructive interference between the spin packets of different offset-frequencies leads to a reduction in echo intensity at higher  $Q_{crit}$  values. This effect is known and discussed in detail by Jeschke et al. (2015), as referenced in line 225-226 (232-233 in revised version). In the revised version, we will include an additional explanatory sentence to clarify this effect to the reader.

Point 2: Further, what is the direct consequence of the increasing B1-inhomogeneity, which causes the observed echo decline at high Qcrit? Is it a more severe phase roll delta\_phi? This is assuming that the fixed echo phases phi\_0 have been corrected for each Qcrit. Is that the case?

Yes the  $\phi_0$  resulting from the sum of all individual B1-inhomogeneity simulations was always refocused for each  $Q_{crit}$ . Every curve in figure 3 (a) represents a sum of multiple simulations where the B1-strength of the pulses was scaled by the B1-strengths and weighted by the B1-weights shown in the inlet (this explained in detail in section 2.2.1). Here we explicitly do not phase  $\phi_0$  for every B1-simulation before summation, since in a realistic measurement the B1 inhomogeneity over the sample volume will cause a distribution of  $\phi_0$  and  $\Delta\phi$  values over the sample. This is ultimately what causes the decline in echo intensity, consequentially if the B1-inhomogeneity is more extreme the larger variation in  $\phi_0$  and  $\Delta\phi$  will cause a faster decline in echo intensity with  $Q_{crit}$ .

Your question prompted us to try to investigate which of the two phase shifts ( $\Delta \phi$  and  $\phi_0$ ) is the dominant source of the decline in Echo intensity with increasing  $Q_{crit}$ . For this we

have repeated the simulations in Figure 3, but in one case selectively phased  $\phi_0$  (not  $\Delta\phi$ ) of the individual B1-inhomogeneity simulations before summation and in the other case tried to refocus  $\Delta\phi$  but not  $\phi_0$  before summation.

The results are shown in the following figures:

Here it is evident that the distribution of  $\phi_0$  is the dominant effect. While in absence of B1-inhomogeneity (pink trace) only  $\Delta\phi$  causes a decline in echo intensity as discussed in the previous question. This highlights the importance of considering  $\phi_0$  and we have included these simulations in the appendix of the revised version. This observation is also consistent with our observations in Figure 1 and Figure 4. In Figure 1 we had already shown that  $\phi_0$  is large compared to  $\Delta\phi$  (Line 200 (207 in revised version)) and that  $\phi_0$  depends on pulse length but  $\Delta\phi$  does not (as long as all spins are inside the excitation bandwidth of the pulses). The different declines of echo-intensity in Figure 4 dependent on pulse length must therefore also be caused by  $\phi_0$ .

Point 3: In the text (p. 10) it should be "small sample", rather than "large sample" (the stated color is correct).

Thank you for pointing this out! However, the label "large sample" is in fact correct, while the stated color is incorrect. We will correct the color assignment in the revised version.

In relation to the nice simulations in Fig. 4, p. 10, the term "destructive interference pattern" is unclear. Also, please confirm/state in the text clearly that all sequences in Fig. 4 are set up in a way that complies with Eq. (8), as briefly indicated in the figure caption.

**Response:**

With the destructive interference we refer to the fact that different parts of the sample which experience a different B1-field strength (due to B1-inhomogeneity), will experience different dynamic phase shifts and therefore their individual magnetization components will destructively interfere as soon as phase shifts become large. We assume that you are referring to line 246 (257 in revised version). We believe the sentence is already clear but changed it to:

"Since this destructive interference is caused by a distribution of different dynamic phase shifts ( $\Delta \phi$  and  $\phi_0$ ), the effect is much less pronounced when.."

We have also included a sentence in the text of the revised version to clearly state that all simulations in Fig. 4 are set up to refocus the parabolic phase shift  $\phi_p$  according to Eq. (8) (line 252 in revised version).

In line with the comment of Nino Wili, it should be correctly explained that working in the linear regime of the microwave amplifiers is not a requirement, contrarily to the authors' statement. It should be discussed that amplifier non-linearity or compression should be quantified and accounted for when calculating the microwave pulse shapes, or otherwise (when on purpose avoided for model studies) significant microwave power needs to be sacrificed to remain in the linear regime of a TWT amplifier. To avoid the caveat of sacrificing microwave power, software compensation with experimental amplitude compression appears preferable.

**See comment above**

For the instrumentation, in the absence of high-speed clocks, it would be better to report a jitter time for single echoes (1 shot) for the standard and the high-speed detection variants because without high-speed clock synchronization there is also no fixed phase relation. I strongly suspect, that for down-conversion to 0 GHz before the ADC a small remaining jitter should have indeed only marginal influence (if any) – however, chirp echoes will include higher frequency components, hence jitter should be known before averaging shots to quantify echo amplitudes.

See the comments above, we observe an average jitter time of approximately 0.2 ns. This does not cause any significant attenuation at higher frequencies, as is also evident by our very good agreement between the spectrum obtained by FT and by echo detected field sweeps.

In agreement with the public comment of Maxim Yulikov, I share his opinion that the discussion of the SIFTER background and particularly of its correction here falls short of the potential offered by the presented data. It is a great first step that experimental SIDRE traces have been obtained, and surely also a 2-pulse Hahn echo decay should be known for these samples. Now that a strategy for background correction is published, as cited by Vanas et al 2023, it appears adequate to at least discuss why the presented approximation was chosen and explain how this could be justified over obtaining the SIFTER form factor F(t) with  $F(t) = [S\_SIFTER(t) - S\_unmodulated(t)] / [S\_SIDRE(t) + S\_2Hahn(t)]$ , where S are the different time domain signals and  $S\_2Hahn(t)$  is the decay of 2 Hahn echoes multiplied back-to-back as described by Vanas et al 2023. The approach taken here and the resulting quantitative agreement with the PELDOR data, suggests that the  $S\_2Hahn(t)$  signals here may have a form very similar to the  $S\_SIDRE$  signal for the length of the SIFTER traces shown here. These points appear important to test and clarify in order to provide solid grounds for the background correction employed here.

**Response:**

First we want to state, that our approach with just dividing by the SIDRE trace already compares very well with the PELDOR form factors measured independently and analyzed as usual (see new Figure C4c). The exponentials (with very long time constants) were only introduced to remove an artifact in the 2D FT spectra at zero frequency (and did not affect the peaks at the dipolar frequency). That this leads to a quantitative agreement with the form factors obtained by PELDOR is probably due to several reasons: the unmodulated

part of the SIFTER signal is very small in our case with broadband pulses and there seems to be no T2 filtering effect in our samples. This might be due to the fact that our rigid spin labels are oriented towards the center of the RNA helix and therefore all see exactly the same nuclear spin surrounding.

Secondly, we had problems to apply the procedure described in Vanas et al. 2023 to our data. Our nonselective chirped 2-pulse Hahn echo decays have strong dipolar modulations so an accurate Hahn echo decay without modulation is very difficult to obtain. Even in Hahn echo decays with fairly long rectangular pulses (16-32ns) pronounced dipolar oscillations are visible. Additionally, we found slightly different decay curves, depending on the pulse length, which again lead to some uncertainty in the analysis. There was also some uncertainty to fit the very small unmodulated part of the SIFTER trace before subtraction. Indeed, here a fit with a stretched exponential lead to better agreement with the PELDOR time traces than a fit with the exact formula given in Vanas at al. 2023.

For this reason, we used our much simpler approach to obtain the SIFTER form factors here. In the revised manuscript we describe this in more detail.

While indeed Chirp echoes are not the current experimental standard, the literature is somewhat less sparse than it sounds in the introduction. In addition to the works already cited, there are a number of notable applications optimizing and using chirp echoes, such as ESEEM (DOI 10.1063/1.4927088), chirp RIDME (doi:10.2533/chimia.2019.268), and CHEESY-detected EDNMR (DOI 10.1016/j.jmr.2018.02.001 & DOI: 10.1039/D0SC04436A). Also, experimental characterization of pulse excitation profiles has been shown and used as calibration (e.g. Fig. 6 b/e in DOI: 10.1039/c7cp01488k).

**Response:**

Thank you. We included the citations to the introduction.

In the discussion of Fig. 2 it could be made clear that the sequence with pulse length ratios of 2-2-2 does not satisfy Eq.(8), while the ratio of 2-2-1 does. The expected result is the pronounced phase roll observed in panel (b) (for which the fig. captions is incomplete). The expectation based on Eq. (8) together with this result could be discussed more clearly.

**Response:**

We do believe that this is made clear in line 214-216 (223-24 in revised) ("...and in the second case  $\phi_p$  is not refocused..."). As the phase roll  $\phi_p$  that we defined in section 2.2.2 is exactly this,  $\phi_p$  is the phase roll that is refocused by Eq. (8). We also believe that the expectation based on Eq.8 is clearly discussed in line 218-220 (225-226 in revised version). We added a reference to equation 8 in line 211 (218 in revised version) where we mention the parabolic phase roll: "To refocus the parabolic phase roll  $\phi_p$  (see Eq.8)..."

p2 When discussing instrumental non-linearities, Ilya Kuprov's GRAPE paper (doi: 10.1063/5.0264092) could be cited as solution here to be aware of and discuss the approach to compensate for hardware imperfections in software.

**Response:**

Thank you we included the citation.

p2 When other signals (here called unwanted) of SIFTER or DQC are discussed, it should be noted that most other coherence transfer pathways are not deleterious anymore, but instead can be explicitly taken into account during data analysis in DeerLab using multipathway fitting.

**Response:**

We will mention this possibility in line 50 (54 in revised version). Nevertheless, we do believe that incomplete excitation is detrimental to chirp-SIFTER and cannot be compensated by fitting of additional pathways. Incomplete excitation will not just cause artefacts which under introduction of additional fitting parameters (increasing uncertainty) can be fitted, but will also cause loss in modulation amplitude and therefore in SNR.

p3 As mentioned (and known, see e.g. Verstraete et al 2021) there is a difference between the nominal setting and the actual effective excitation bandwidth of chirp pulses. How much is that difference under the conditions used here? This is e.g. experimentally quantified in pulse excitation profiles.

**Response:**

The pulse sweep width was 500 MHz and the excitation bandwidth approximately 240 MHz. Slightly different for the different pulse lengths used (all details given in Appendix A3). We have verified this with the profiles shown in figure 8 (c) and in more detail in figure A5 of the Appendix.

p4 The finding that smaller Qcrit values (with large TBP values) for pi-pulses can lead to larger echo amplitudes is also discussed in the Verstraete 2021 paper. With the minimum value of TBP = 30 for pi-pulses, it would be helpful to also mention suitable values for the pi/2-pulses used here.

**Response:**

Actually, we show the opposite in Figure 4. For pulse sequences with an uneven number of pi pulses under B1-inhomogeneity it is best to have low TBPs because of the less steep slope of  $\phi_0(Q_{crit})$  (still of course a minimum of around TBP $\approx$ 30 is required to achieve homogeneous excitation). We believe that it is in general not beneficial to give a minimum value for the TBP, the excitation profile of the pulses, both the  $\pi/2$  and  $\pi$  pulse should always be simulated in advance to determine if the excitation bandwidth is large enough. We have written this also in line 112-113 (118-119 in revised version). Giving default values can lead to mistakes as the excitation bandwidth is dependent on all pulse parameters. All our SWs, pulse lengths and WURST truncation parameters are given in section A3 of the appendix.

p6 While defining the different phases, please also define phi\_p the parabolic phase shift, and for clarity please confirm that within your definitions that the Bloch Siegert shift and the term "dynamic phase shifts" are to be considered equal.

**Response:**

We have defined the "parabolic phase shift"  $\phi_p$  (refocused with Eq.8) in section 2.2.2 and referred to equation 8 in line 172 (179 in revised version) of section 2.2.3. The term "dynamic phase shifts" was introduced by Jeschke et al. 2015 to refer to the fact that they are dependent on  $Q_{crit}$ . As already cited by us (line 173 (181 in revised version)) these effects can be tracked back to Bloch-Siegert phase shifts (Emsely and Bodenhausen, 1990).

e.g. p9: Please don't omit "chirp" in 2-pulse chirp Hahn echo for clarity, as many might understand the conventional Hahn echo to have rectangular pulses.

**Response:**

Thanks, we have modified the revised manuscript accordingly.

p12 and appendix: What is an AWG input amplitude, considering that the AWG is a microwave source?

**Response:**

Our AWG has a 14 Bit resolution in amplitude. The input amplitude refers to this bit resolution. Using the full range would be 100 %, half the range 50 % and so on.

p16, Fig. 8a it would be nice to mark the 0 MHz offset point in the resonator profile for the other panels in Fig 8.

**Response:**

Thanks, we have modified the revised manuscript accordingly.

p17 In the brief introduction of orientation-selective PELDOR, particularly here with respect to X-band, also DOI: 10.1016/j.jmr.2011.12.024 would be adequate to cite.

**Response:**

Thanks, we have modified the revised manuscript accordingly.

p19 "SIFTER does not have these limitations and the frequency resolution is only limited by the homogeneous linewidth and the SNR..." Why homogeneous rather than inhomogeneous linewidth here? The EPR spectrum consists of a large number of inhomogeneously broadened EPR lines, some of which are excited at each frequency step during a chirp pulse.

**Response:**

Thank you for your comment. Technically, the frequency resolution in the 2D experiment in the spectral dimension is given by 1/(length of the echo time trace) recorded. Of course, this is artificial because the echo signal decays much earlier. Therefore, you are correct, the resolution in this frequency dimension at X-band is dominated by inhomogeneous Gaussian line-broadening resulting from unresolved proton hyperfine couplings. The wording "is only limited by the inhomogeneous linewidth" might nevertheless lead to some confusion, since commonly inhomogeneous linewidth of a nitroxide powder spectra refers to the full spectral width dominated by the anisotropic nitroxide hyperfine tensor. We therefore propose the following wording:

"SIFTER does not have these limitations and the frequency resolution at X-band is only limited by the additional inhomogeneous line-broadening arising from proton hyperfine couplings and the SNR of the recorded data"

p19 ".. dipolar frequencies from omega\_dd to 2 omega\_dd at the edges of the spectrum." Fig. 10 a looks like omega\_dd is observed at the two edges, whereas 2 omega\_dd is observed in the center (around 0 MHz offset in Fig. 10).

**Response:**

This is not the case and our wording is correct. For clarification also see figure 11 (b).

p26, Fig. 15 If a SIDRE trace is available as reference to the 6-pulse SIFTER, it would be nice to also show this in Fig. 15.

**Response:**

We have included the 5-pulse equivalent of the SIDRE trace in the appendix in figure C6 as referenced in the main text in line 529 (line 552 in revised version).